# PROTMAMBA: A HOMOLOGY-AWARE BUT ALIGNMENT-FREE PROTEIN STATE SPACE MODEL

## ABSTRACT

Protein design has important implications for drug discovery, personalized medicine, and biotechnology. Models based on multiple sequence alignments efficiently capture the evolutionary information in homologous protein sequences, but multiple sequence alignment construction is imperfect. We present ProtMamba, a homology-aware but alignment-free protein language model based on the Mamba architecture. In contrast with attention-based models, ProtMamba efficiently handles very long context, comprising hundreds of protein sequences. We train ProtMamba on a large dataset of concatenated homologous sequences, using two GPUs. We combine autoregressive modeling and masked language modeling through a fill-in-the-middle training objective. This makes the model adapted to various protein design applications. We demonstrate ProtMamba's usefulness for the generation of novel sequences and for fitness prediction. ProtMamba reaches competitive performance with other protein language models despite its smaller size, which sheds light on the importance of long-context conditioning.

## 1 INTRODUCTION

Proteins are essential building blocks of life, serving vital roles in metabolic processes, cellular transport, structural integrity, and immune responses. Composed of long chains of amino acids (polypeptides), proteins fold into specific three-dimensional structures critical for their biological functions. One of the key challenges in biology is protein engineering and design: conceiving protein sequences to exhibit enhanced or novel functions. While experimental approaches like directed evolution and mutational scanning are effective in this regard, they only allow exploring the neighbors of existing sequences. However, the recent growth of extensive databases has opened up new avenues for computational methods that exploit the breadth of biological evolution. For instance, UniProt (The UniProt Consortium, 2021) contains more than two hundreds of millions of protein sequences. Biological functions exert evolutionary constraints on protein sequences, which can be probed by considering families of homologous proteins (i.e. proteins that share an evolutionary history) and analyzing this data through statistical methods and, more recently, through deep learning methods.

Protein language models rely on recurrent (Bepler & Berger, 2019), transformer (Rives et al., 2021) or convolutional (Yang et al., 2024) architectures, and are trained through masked language modeling, autoregressive modeling, or discrete diffusion techniques (Alamdari et al., 2023), on large ensembles of single protein sequences (Khakzad et al., 2023). The representations learned by these models correlate with biochemical properties of proteins (such as function, structure, contacts) (Elnaggar et al., 2021; Vig et al., 2021; Rives et al., 2021; Madani et al., 2023), and can be used to generate protein sequences or to evaluate the fitness of variants. The vast majority of these methods are trained on non-structured ensembles of single protein sequence and do not have direct access to homology, or to conservation and variability within protein families. Models trained on multiple sequence alignments (MSAs) of homologous sequences have also been introduced, despite raising memory challenges and potentially suffering from the imperfections of MSAs (Thompson et al., 2011). Successful MSA-based transformer models, such as MSA Transformer (Rao et al., 2021) or the EvoFormer module of AlphaFold2 (Jumper et al., 2021) alternate attention along protein sequences and across homologs. More recently, PoET (Truong Jr & Bepler, 2024) was trained on concatenations of non-aligned homologous sequences, offering a promising autoregressive alternative to MSA Transformer for protein fitness prediction and design.

State space models such as S4 (Gu et al., 2021), Hyena (Poli et al., 2023) and Mamba (Gu & Dao, 2023) are catching up with transformers thanks to their ability to efficiently handle very long sequences of tokens. These models were quickly adapted to work with biological data. Approaches such as HyenaDNA (Nguyen et al., 2023) or Evo (Nguyen et al., 2024) were trained on long DNA sequences and capture regulatory mechanics. Meanwhile, PTM-Mamba addresses post-translational modifications of protein sequences (Peng et al., 2024).

In this paper, we present ProtMamba, a novel homology-aware but alignment-free protein language model, trained on concatenated sequences of homologous proteins. Based on the Mamba architecture (Gu & Dao, 2023), ProtMamba is able to handle extremely long contexts (unlimited lengths during inference). Trained to autoregressively predict the next amino acid, but also with a fill-in-the-middle (FIM) objective, it can be used for multiple different tasks. First, ProtMamba can autoregressively generate novel sequences without contextual information. Second, by providing ProtMamba with sequences from a specific protein family or subfamily as context, users can prompt it to generate sequences tailored to their specifications. This conditional generation approach is a key strength of the model (see also Truong Jr & Bepler (2024)), and could become an alternative to fine-tuning. Third, ProtMamba supports sequence inpainting, i.e., filling specific masked regions with the desired number of amino acids. For this, along with homologous sequences (used as context), the model is provided with a target sequence to be modified. This generation mode opens novel methods of designing specific parts of protein sequences. Furthermore, ProtMamba is useful for fitness prediction tasks. Users can input a sequence with specific masked positions, prompting the model to output the probability distribution of all mutations in each variant with a single forward pass. Across these various tasks, we obtain competitive results with larger protein language models and task-specific methods.

## 2 METHODS

### 2.1 KEY TECHNICAL CONTRIBUTIONS

1. To harness the evolutionary information present in homologous sequences without relying on multiple sequence alignments (MSAs), we use as input a concatenation of homologous sequences for each protein family. In each of these long arrays, sequences are separated with a specific token. The motivation is that evolutionary information is extremely useful for protein modeling Jumper et al. (2021); Rao et al. (2021); Abramson et al. (2024), but MSAs can be inaccurate. This approach is similar to that used recently in the autoregressive transformer PoET (Truong Jr & Bepler, 2024).

2. We develop an architecture based on Mamba blocks, an alternative to attention recently proposed by (Gu & Dao, 2023) that relies on state space models. In Mamba, which is a recurrent neural network, time complexity scales linearly in sequence length, bypassing the quadratic time complexity constraints of transformers. This allows handling significantly longer input sequences, in addition to being faster to train and to use at inference. This is a key asset here, as concatenating homologous sequences results in long inputs. Note that Truong Jr & Bepler (2024) employed attention matrix chunking to address this issue, but this results in potential losses of statistical dependence signals, and only partially solves the memory limit.

3. We combine elements of both autoregressive modeling and masked language modeling (MLM), by training our model using the fill-in-the-middle (FIM) objective (Bavarian et al., 2022; Fried et al., 2022; Raffel et al., 2020). The model learns to predict masked patches extracted randomly from a sequence and positioned at the end of it, and can therefore leverage the full sequence context, while being trained autoregressively. This is of particular interest for biological sequences, because preceding and subsequent tokens can all be informative to predict a new token. While autoregressive models are generative by definition, they yield the probability of each new token conditioned on previous ones (ignoring subsequent ones). Besides, MLM can be productively used for protein sequence generation (Sgarbossa et al., 2023).

4. To promote the model's ability to reason over in-sequence positions, which is particularly useful for the FIM task, we modify the original Mamba implementation by introducing sequence-level positional embeddings. This enables the model to pay attention to relative

positions inside each sequence. In inference and generation, it opens the possibility of controlling the number of amino-acids to generate.

## 2.2 MODEL ARCHITECTURE AND TRAINING STRATEGY

ProtMamba's architecture is adapted from Mamba (Gu & Dao, 2023). An important modification is that we introduce learned positional embeddings for the input tokens. Among different variants (see supplementary section C), we observed that the most effective and stable method to integrate positional embeddings is to concatenate them with the input token embeddings into a single vector. Specifically, we allocated half of the embedding dimension $d$ to token information and the other half to positional information.

We trained a 107 million parameters model with 16 layers, embedding dimension $d = 1024$, and hidden state dimension equal to embedding dimension. We started with a maximal total input sequence length of $2^{11} = 2048$ amino acids (recall that input sequences are concatenated homologous protein sequences). The model was trained following (Gu & Dao, 2023) with some minor modifications. We used the AdamW optimizer with the following parameter values: weight decay $w = 0.1$ and $(\beta_1, \beta_2) = (0.9, 0.95)$. We scheduled the learning rate to increase from zero to $6 \times 10^{-4}$ with a linear warm-up of $500$ steps followed by a constant learning rate. To optimize memory usage, we trained the model using the `bfloat16` format.

To avoid training instabilities observed in (Nguyen et al., 2024; Waleffe et al., 2024), we implemented a callback mechanism to revert to a previous checkpoint if the loss never assumed values below a threshold for 10 successive evaluation steps. The threshold value was chosen as the lowest training loss increased by $0.5\%$. This ensures that the loss decreases overall, while allowing it to transiently increase. We also prevented gradient explosions by clipping the gradient norm to $1.0$.

The model was trained by scheduling the context length of the input using sequence length warm up (SLW) (Nguyen et al., 2023). Initially, we used inputs of length $L = 2^{11}$ tokens with a batch size of $64$. We doubled input length each time the loss reached a plateau, simultaneously reducing batch size to maintain a fixed total number of tokens per batch. In case of memory constraints, we decrease the batch size and use gradient accumulation. This heuristic approach is based on the idea that a longer context should provide more information. It is useful because of training instabilities for long contexts (Nguyen et al., 2023; 2024). Note that we did not start training the model with a long context to benefit from a larger batch size, which helps to approximate the loss landscape more efficiently. Finally, once we reached a context length of $L = 2^{17}$, we implemented gradient checkpointing to minimize memory consumption. This allowed us to increase the batch size for the final part of the training and obtain a better approximation of the loss landscape, see (Nguyen et al., 2023; 2024).

The model was trained on one NVIDIA RTX A6000 GPU for 35 days, and then on two of them for 15 days. This allowed us to keep the batch size large enough when the context size increased. In total, the model was trained on $1.95 \times 10^{11}$ tokens (approximately 1.5 epochs) and used $2.0 \times 10^{20}$ FLOPs during training. These numbers show the huge improvements that the Mamba architecture has in terms of training speed with respect to transformers. As a comparison, the smallest ESM3 model (Hayes et al., 2024) was trained with $0.8 \times 10^{11}$ tokens using $6.72 \times 10^{20}$ FLOPs, which means that given a fixed amount of compute, ProtMamba can see 8.5 times the tokens seen by ESM3. See Figure S1 for the training curves.

We consider two different ProtMamba versions that were obtained by saving checkpoints at different moments of the training. Our model *ProtMamba, Foundation* was trained on a maximum context length of $2^{15}$ tokens. Our model *ProtMamba Long, Foundation* was trained until the context length reached $2^{17}$ tokens. Both models were fine-tuned for 2 days on predicting only the FIM amino acids to improve inpainting capabilities, yielding the models *ProtMamba/ProtMamba Long, Fine-tuned*.

We also performed multiple ablations on the model architecture and on the different modalities by training models on 10B tokens for 50k steps (see supplementary section C).

## 2.3 DATASET CONSTRUCTION

We trained ProtMamba on OpenProteinSet (Ahdritz et al., 2024), a dataset which comprises 16 millions MSAs, one for each sequence cluster within Uniclust30 (Mirdita et al., 2017). This dataset

was curated to train OpenFold (Ahdritz et al., 2022). We used a filtered subset of the full dataset, consisting of maximally diverse representative MSA clusters, built by iteratively eliminating redundant clusters whose representative sequences appeared in other clusters' MSAs (Ahdritz et al., 2024). This ensures that each representative sequence is only present in its cluster, as detailed in (Ahdritz et al., 2024). This dataset comprises 268,000 clusters including a total of 508 million sequences and 110 billion residues (see Figure S2 for additional statistics). A validation set and a testing set are formed by holding out respectively 192 and 500 randomly chosen clusters from the training set. Importantly, our use of the filtered version of OpenProteinSet (Ahdritz et al., 2024) ensures that overlap between clusters in the training, validation and test set is strongly minimized. Indeed, this filtering is based on selecting only MSAs of maximal diversity and ensuring that the reference sequences used to build each cluster are not present in any other cluster.

Figure 1 illustrates the construction of a training example. First, a cluster is randomly selected from the filtered OpenProteinSet database described in Section 2.3. As OpenProteinSet uses MSAs, we restore the original unaligned sequences by removing gaps and converting all lowercase insertion residues to uppercase. Each amino acid is tokenized using a unique token. Then, $N$ sequences are sampled uniformly at random and concatenated into a single array, with a **<cls>** token separating each sequence from the next one. The value of $N$ is chosen for the total length of the concatenated sequence to exceed the desired training context length $L$ (e.g. $L = 2^{11}$ at the beginning of training), and the input is then cropped precisely at $L$. Next, the sequences are prepared for the FIM task. For each sequence, some patches of consecutive tokens are randomly sampled (see below) and masked by replacing them with a mask token **<mask i>**, with one such token representing patch $i$. For each patch, we append to the sequence another mask token followed by the corresponding masked amino acids (which are unmasked). An **<eos>** token is used to separate the main (masked) sequence from its unmasked patches.

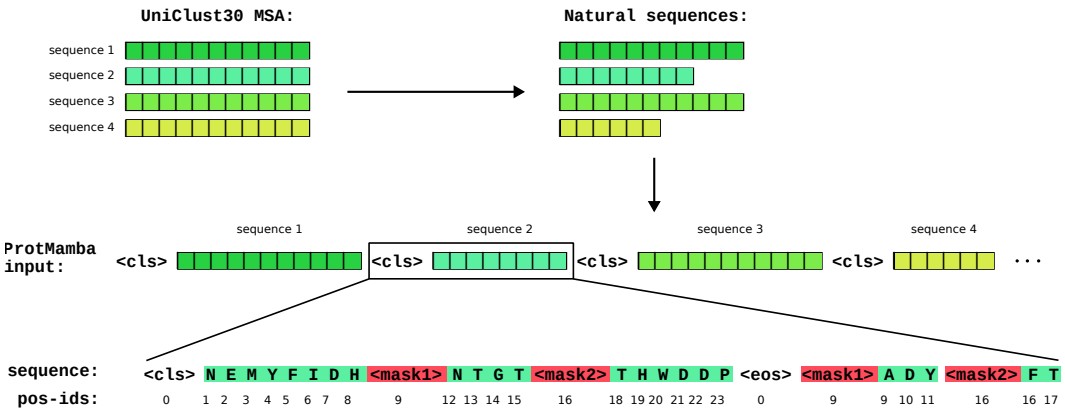

Figure 1: **Input to ProtMamba.** Each element of the input is a concatenation of unaligned homologous sequences separated by **<cls>** tokens. Each sequence starts with a **<cls>** token and ends with an **<eos>** token. Masked segments are replaced by numbered mask tokens, **<mask1>**, ..., **<mask5>**. The masked tokens are appended to the sequence, after the **<eos>** token, each masked segment being preceded by its associated mask token. The position indices ("pos-ids") follow the succession of tokens in the natural sequence. Thus, the masked tokens have their initial position indices in the natural sequence. The position index of each mask is set to that of the first associated masked token. In this particular example we sampled two masks $i = 1, 2$ with length $P_1 = 3$ and $P_2 = 2$.

The following rules are applied when masking each sequence:

1. The number of masked patches in a sequence is sampled from a Poisson distribution with $\lambda = 1$, and capped at 5 (by resampling in case values above 5 are obtained). This yields no mask in $36\%$ of sequences, one mask in $36\%$ of sequences, and more in $28\%$ of sequences.
2. The starting position of each patch is sampled uniformly (without replacement) from all possible positions in the sequence.
3. The length $P_i$ of each patch $i$ is sampled uniformly in $[1, \max(P_i)]$, where $\max(P_i)$ is 0.2 times the distance from the start point of patch $i$ to the start point of patch $i + 1$ (or to the end of the sequence for the last patch). This ensures that no more than 20% of all tokens

in each sequence are masked, in line with masking fractions of similar models (Rao et al., 2021; Rives et al., 2021).

Finally, each token is allocated a position index (used to obtain the associated positional embedding) that tracks its position in the original sequence. The position indices of **<cls>** and **<eos>** are set to zero, while the mask tokens **<mask i>** have the same position indices as the first token they are masking, see Figure 1.

## 3 RESULTS

### 3.1 PROTMAMBA BENEFITS FROM LONG CONTEXT

To evaluate the effectiveness of incorporating context information in ProtMamba, we examine the scaling of the model's perplexity with context length for natural sequences. Perplexity is commonly used to evaluate autoregressive models and assesses how uncertain they are about a sequence. It is the exponential of the cross entropy loss. Figure 2 shows the scaling of perplexity for the masked parts of the sequences as a function of the number of context sequences, when using the FIM objective. ProtMamba Long (Fine-tuned) achieves remarkably low values of perplexity for small numbers $N_m$ of masked tokens. Furthermore, perplexity decreases when increasing the number of context sequences, revealing the positive impact of richer context on model performance. This decrease tends to be steeper for larger $N_m$, suggesting that these difficult tasks particularly benefit from richer context. Given the diverse lengths of sequences across protein families, we report perplexity versus the number of sequences in the context rather than versus the total length of the context. Indeed, there can be different amounts of information in contexts of similar lengths but composed of sequences of varying lengths.

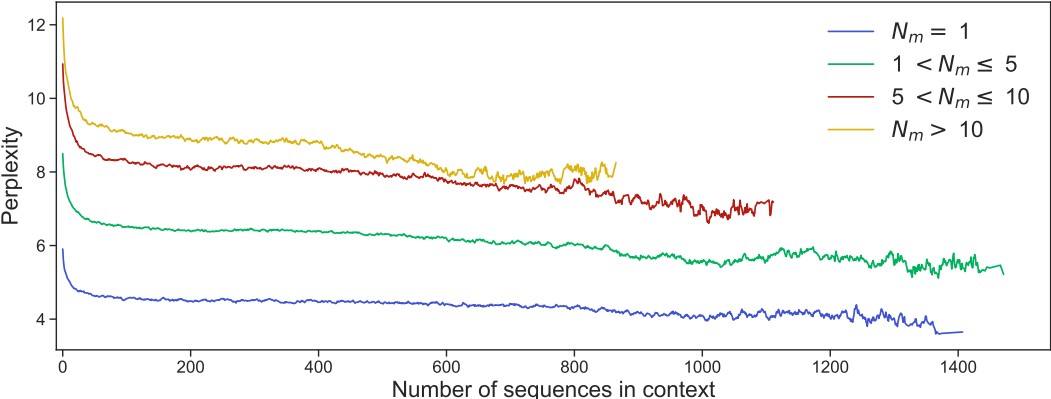

Figure 2: **Scaling of the FIM perplexity with the number of context sequences.** We show the FIM perplexity for different numbers $N_m$ of masked amino acids versus the number of context sequences. Results are averaged over all 500 clusters of the test set and 100 replicates for each cluster (differing by the random sampling of context sequences). Context sizes go up to $2^{17}$ amino acids. To reduce noise, we take the exponential moving average, and we restrict to cases where the count of samples is at least 100. See Figure S3 for a log-log version of this figure.

Furthermore, we study the scaling of the per-sequence perplexity (i.e. the standard autoregressive perplexity of the full non-masked sequence) computed on the test set using ProtMamba Long (Foundation), see Figure S4. Initially, we notice a decrease of perplexity to a minimum of 7.70 as the number of sequences in the context increases, with lower perplexity values for shorter individual sequences, but this reduction plateaus after a certain point. We attribute this behavior to the finite size ($d = 1024$, see Section 2) of the hidden state of the model, which limits its capacity to effectively leverage context information at each step. We hypothesize that a larger model with a higher-dimensional hidden state could increase the amount of information transferred from the context to the next predicted token. For completeness, we also report perplexity versus context length measured in tokens (see Figure S5). There, we observe a rise in perplexity when the context sizes reaches $2^{17} = 131,072$ tokens, which is the highest context length seen during training. We expect

that further training the model for longer contexts could lead to lower perplexity values, yet ultimately reaching a lower bound due to the limitations imposed by the hidden state dimension and model size.

## 3.2 PROTMAMBA PREDICTS MUTATIONAL EFFECTS IN DIFFERENT PROTEIN FAMILIES

Next, we evaluate ProtMamba's ability to predict mutational effects, leveraging its inpainting capabilities arising from the FIM training objective. Indeed, by masking specific amino acids in the wild-type sequence of interest, we can predict the fitness of all variants at these sites. Our first step to evaluate variant fitness is to collect a context of homologs to the wild-type sequence. We use the ColabFold protocol (Mirdita et al., 2022) for this, ensuring that diverse sequences are found in a few minutes. Then, we randomly subsample 200 sequences among those that have between 30% and 98% similarity to the wild type to construct the context, and we sort these sequences by increasing similarity to the wild type, as in Truong Jr & Bepler (2024).

To evaluate the effect of a variant with a single mutated site, we append the wild-type sequence to the context, mask the mutated residue in it, and predict this residue using the FIM method. Let $\mathcal{C}$ denote the union of the context sequences and of the wild-type sequence masked at the mutated position $i$. We evaluate the effect of mutations at position $i$ by their fitness score $\mathcal{F}$, defined as:

$$\mathcal{F}(i, x_i, \mathcal{C}) = \log p(x_i|\mathcal{C}) - \log p(x_i^{WT}|\mathcal{C}),$$

for all residues $x_i$ different from the wild-type residue $x_i^{WT}$. Using this method based on the FIM objective allows us to evaluate the effects of all mutations at position $i$, decreasing 20-fold the number of passes through the model needed to evaluate all mutations with respect to the typical method used with autoregressive protein language models. To predict the fitness effects of variants involving mutations at multiple sites, we add all the single mutation likelihoods. This approximate, but fast method avoids computing the complete likelihood for all variants, thus reducing the number of calls to ProtMamba. It is accurate when the mutations can be considered independent. We also test variant scoring by ProtMamba using the autoregressive log-likelihood loss instead of the FIM loss (this approach is called "ProtMamba AR"). Details on the different approaches we employed to predict mutational effects with ProtMamba are given in Supplementary section B.

We consider the ProteinGym benchmark (Notin et al., 2023), which contains 217 datasets of substitutions in protein sequences (both single and multiple) and allows comparing to state-of-the-art methods. In Table 1, we report the performance of ProtMamba, and we compare it to published models classified by type: alignment-based models, single sequence protein language models (PLMs), aligment-enhanced PLMs, homology-aware PLMs, and structure-aware models. Table 1 shows that variant scoring by ProtMamba using FIM outperforms using the autoregressive log-likelihood (ProtMamba AR). All ProtMamba performances reported in Table 1 are those of ProtMamba Long fine-tuned on the FIM task, except for ProtMamba AR where we used ProtMamba Long. Indeed, considering the 4 different ProtMamba versions (see Section 2), we found that ProtMamba models fine-tuned on the FIM task outperform foundation models, and that ProtMamba Long performs better than ProtMamba, confirming the importance of training the model with a long context (see Figure S6). In Figure S7(a), we break down the performance of ProtMamba Long for different context lengths and different protein sequences lengths. We observe that variants with long sequences particularly benefit from long contexts, as they allow including more sequences. This interpretation is supported by Figure S7(b), which shows that this dependence on context length is weaker when considering context length in terms of number of sequences. Based on performance on a validation set (see supplementary Section A and Figure S8), we chose to use a context of 200 sequences to predict fitness using ProtMamba Long (fine-tuned).

Table 1 shows that ProtMamba outperforms single-sequence PLMs ("PLM" type) of the same size (ESM-2, 150M), and performs similarly or better than larger models like Tranception L and ESM-2 (650M). This illustrates the power of homology information for mutational effect prediction.

Since MSA information remains useful in scoring variants, in the rows "Alignment + PLM" of Table 1, we show results where explicit use of MSAs was made, either via retrieval, i.e. ensembling the models with an independent-site model, as in Notin et al. (2023) (denoted by "R"), or by combining a PLM with GEMME in Marquet et al. (2024). Using retrieval, ProtMamba (w/ R) obtains similar performance as Tranception L (w/ R) and as MSA Transformer, which leverage MSA information. These two models were trained using more than one order of magnitude more FLOPs than ProtMamba.

| Model type | Model | #params | $\rho$ | Time | Citation |
|---|---|---|---|---|---|
| Alignment-based | Site-Independent | - | 0.359 | - | Hopf et al. (2017) |
| | GEMME | - | 0.455 | - | Laine et al. (2019) |
| PLM | Tranception L (w/o R) | 700M | 0.374 | - | Notin et al. (2022) |
| | ESM-2 | 150M | 0.387 | - | Lin et al. (2023) |
| | ESM-2 | 650M | 0.414 | - | Lin et al. (2023) |
| Homology-aware | **ProtMamba** (single) | 107M | 0.406 | 7m | |
| PLM | **ProtMamba AR** (single) | 107M | 0.367 | 1h 39m | |
| | PoET (single) | 201M | 0.447 | 9h 51m | Truong Jr & Bepler (2024) |
| | PoET (ensemble) | 201M | 0.470 | 148h* | Truong Jr & Bepler (2024) |
| Alignment | **ProtMamba** (w/ R) | 107M | 0.432 | 10m | |
| + PLM | MSA-Transformer | 100M | 0.421 | - | Rao et al. (2021) |
| | Tranception L (w/ R) | 700M | 0.434 | - | Notin et al. (2022) |
| | VespaG | 3B | 0.458 | - | Marquet et al. (2024) |
| Structure-aware | ESM-IF1 | 142M | 0.422 | - | Hsu et al. (2022) |
| | SaProt | 650M | 0.457 | - | Su et al. (2023) |
| | ProSST | 110M | 0.507 | - | Li et al. (2024) |

Table 1: **Performance of ProtMamba and of existing models on the ProteinGym benchmark.** For different models classified by type, and whose numbers of parameters are given, we show Spearman correlation $\rho$ values between predicted and experimentally measured variant effects in ProteinGym. We denote the MSA-augmented methods with retrieval by "(w/ R)". New models introduced here are highlighted in bold. They include ProtMamba with and without retrieval, where variants are scored using the FIM loss, and ProtMamba AR, where they are scored using the autoregressive log-likelihood. Published results were obtained from `https://proteingym.org/`. For ProtMamba and PoET, we also report the time needed to score all variants in ProteinGym (excluding homolog retrieval). Top performances in terms of $\rho$ and time are highlighted in gray.
* Estimated as $15\times$ the time taken by PoET (single).

In Table 1, the state-of-the-art model on the ProteinGym benchmark is ProSST (Li et al., 2024), a structure-aware model. Among structure-agnostic models, the state-of-the-art model is PoET, a homology-aware transformer. ProtMamba reaches a performance which is only slightly lower than PoET. This is notable, as PoET has twice more parameters, and is much slower at scoring variants than ProtMamba. Table 1 shows that ProtMamba can score all ProteinGym variants in ∼7 to 10 minutes on a single NVIDIA RTX A6000 GPU, while PoET takes ∼10 hours for this, and up to ∼6 days in the ensemble mode (the top structure-agnostic method), using the same hardware.

In Table S1, we break down results by MSA depth and by number of mutations. In Figure S9, we further break down the comparisons between models on ProteinGym by category of experiment (panel (a)), taxonomic category (panel (b)) and sequence length (panel (c)). We also show scores for different models on randomly selected example experimental datasets in Figure S10.

### 3.3 PROTMAMBA ACCURATELY PREDICTS THE ACTIVITY OF CHORISMATE MUTASE ENZYMES

Next, we evaluate ProtMamba, and in particular the power of the FIM objective, on a dataset of experimentally tested natural and *in silico* generated sequences from the chorismate mutase family from Russ et al. (2020). Chorismate mutase functions as an enzyme involved in the catalysis of synthesis of amino acids, and is a domain of the bifunctional chorismate mutase/prephenate dehydratase. We use ProtMamba to evaluate the activity of experimentally studied variants of this enzyme. For this, we sample 100 sequences, either randomly among all natural sequences that were experimentally studied, or randomly among the subset of those that were experimentally shown to be active in *Escherichia coli*. For these two types of context, we test three different protocols to predict the activity of the other variants in the dataset of Russ et al. (2020) with ProtMamba. First, we use only the chorismate mutase domains (cropped sequences) as context, and autoregressively evaluate the likelihood of the full sequence ("from left to right"). Second, we use the full sequences (chorismate mutase/prephenate dehydratase) as context and we evaluate the perplexity of the full sequence autoregressively from left to right. Third, we use the full sequences (chorismate mutase/prephenate dehydratase) as context and evaluate the perplexity of the chorismate mutase domain using the FIM objective.

|                        | Domain | Protein | Protein, FIM |
|------------------------|--------|---------|--------------|
| **Published methods**  |        |         |              |
| DCA energy             | 0.41   | -       | -            |
| Logistic Regression    | 0.43   | -       | -            |
| **ProtMamba**          |        |         |              |
| Context: any variant   | 0.41   | 0.44    | 0.46         |
| Context: active variants | 0.50 | 0.52    | **0.53**     |

| # seq. | # residues | $\rho$ |
|--------|------------|--------|
| 0      | 0k         | 0.31   |
| 5      | 1k         | 0.46   |
| 10     | 3k         | 0.48   |
| 20     | 7k         | 0.51   |
| 50     | 19k        | 0.49   |
| 100    | 38k        | 0.53   |
| 200    | 77k        | 0.52   |

Table 2: **Activity prediction of chorismate mutase variants.** (Left) For published methods (Russ et al., 2020), and for ProtMamba with various context types (rows) and protocols (columns, see main text), we report the Spearman correlation $\rho$ between experimental activity and predictions. Associated ROC curves are shown in Figure S11. (Right) Effect of increasing context size (number of sequences and corresponding total number of residues) on Spearman correlation $\rho$ for ProtMamba predictions, using only active variants in context, full domains and FIM.

The left panel of Table 2 provides a comparison of ProtMamba against published methods (Russ et al., 2020) for the two different context types and the three different protocols. We observe that using only active variants in the context consistently improves the predictive power of ProtMamba. With both context types, using full sequences is better than using only domains, and using FIM improves accuracy. Note in addition that using FIM reduces the computation time per variant compared to autoregressively scoring the full sequence.

The right panel of Table 2 shows the impact of context size on the performance of our best ProtMamba-based activity predictor (using only active variants in context, full domains and FIM). The accuracy of this predictor initially strongly increases with context length, and then appears to plateau from a context length around 75 sequences or 28k residues (see also Figure S12 **a**). Thus, context size plays a critical role for these activity predictions. In Figure S12 **b** and **c**, we further compare the perplexity of the variants, when using only active variants as context, and when using both inactive and active variants as context. We observe that the perplexity of inactive variants is often higher when using a context of active variants, indicating a better ability to predict inactivity. Furthermore, the perplexity of active variants is often lower in this case, showing a better ability to predict activity with high-quality context. Additionally, we display the distribution of the perplexity for ProtMamba using FIM and a context composed of active variants, compared to the experimental activity in Figure S13, and the same perplexity versus the model score from Russ et al. (2020) in Figure S14.

### 3.4 PROTMAMBA AUTOREGRESSIVELY GENERATES PROMISING NOVEL SEQUENCES

Finally, we evaluate ProtMamba on the autoregressive generation of novel protein sequences given a context of known homologs, corresponding to members of a given cluster of sequences. We generate sequences from 19 randomly selected clusters in the test set, varying the following parameters: temperature ($T$), top-$k$ number, and top-$p$ fraction, following the approach proposed by (Ferruz et al., 2022). These parameters are commonly employed to control the output of autoregressive models. At each step, top-$k$ limits their output to the top-$k$ most probable tokens, while top-$p$ only includes the top tokens reaching a cumulative probability $p$. Meanwhile, temperature $T$ adjusts the randomness of sampling. Additionally, we vary the number of sequences in the context to assess the impact of different levels of conditioning on the generated sequences. Specifically, for each cluster, we perform generation using context lengths of $n = 10, 100, 500, 1000$ and $N$ sequences, where $N$ is the total number of sequences in the cluster. For each value of $n$, we consider the following $(T, \text{top-}k, \text{top-}p)$ triplets: $(0.8, 10, 0.9)$, $(0.9, 10, 0.95)$, $(1, 10, 0.95)$, $(1, 10, 1)$, $(1, 15, 1)$. We generate 100 sequences for each $(n, T, \text{top-}k, \text{top-}p)$, obtaining a total of 2500 sequences per family. As expected, we observed that the parameters which promote higher sampling variability tend to yield sequences with higher perplexity. Note that sequences with more than 750 amino acids, i.e. longer than the longest natural sequence considered here, were discarded from further analysis. They represented $\sim 5\%$ of the generated sequences.

We compare the sampled sequences (aggregated across all parameter sets mentioned above) with natural sequences from the cluster used as context for generation using various scores evaluating novelty, homology, and structure.

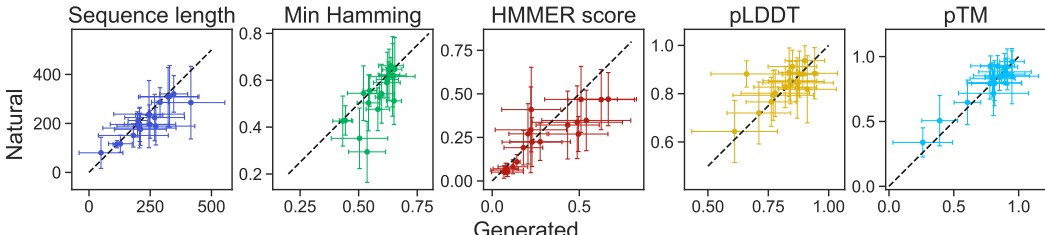

Figure 3: **Comparison of low-perplexity generated sequences with natural ones.** We report the median and the standard deviation of sequence length, Hamming distance to the closest natural neighbor in the sequence cluster from which the context is drawn ("Min Hamming"), HMMER score (rescaled), pLDDT and pTM scores from ESMFold. For each of 19 test clusters, we compare the 100 sequences with lowest perplexity values out of 2500 generated sequences ($x$-axis) with a randomly chosen subset of 100 natural sequences in the sequence cluster ($y$-axis). Dashed black lines: $y = x$.

1. Novelty is assessed by computing the pairwise Hamming distance (using pairwise Smith-Waterman alignment) with each natural sequence in the cluster, after which it is possible to focus on distance to the closest natural neighbor if desired.
2. Homology evaluation involves training an HMM (using HMMER (Eddy, 2020)) on the cluster's MSA, obtained from OpenProteinSet, and computing the scores it gives to generated sequences.
3. Structure is assessed by predicting the structure of each sampled sequence using ESMFold (Lin et al., 2023). As ESMFold is a single sequence model, it provides predictions that are less biased by MSAs than those of MSA-based models. Futhermore, it is faster than AlphaFold2. ESMFold's confidence measures, both global with pTM scores and local with pLDDT scores, allow for a precise comparison of different sequences sampled from the same cluster.

We observe that ProtMamba's estimated sequence perplexity correlates well with HMMER scores, Hamming distance to the closest natural neighbor in the cluster and structural scores (see Figure S15). Thus, ProtMamba assigns lower perplexity values to sequences that are more likely to be part of the cluster. The absolute Pearson correlation value averaged over all clusters and scores is above 0.57. Detailed results for each family and each score are presented in Figure S16. Figure 3 shows that the median scores of our generated sequences that have low perplexity are comparable to those of natural ones. Overall, these results are promising for protein design applications.

| Model | ProtMamba | EvoDiff-MSA | MSA Trans. | Potts | Natural |
|---|---|---|---|---|---|
| pLDDT ($\uparrow$) | **0.75 $\pm$ 0.13** | 0.60 $\pm$ 0.16 | 0.54 $\pm$ 0.18 | 0.56 $\pm$ 0.14 | 0.77 $\pm$ 0.13 |
| scPerplexity ($\downarrow$) | **2.63 $\pm$ 0.45** | 3.17 $\pm$ 0.58 | 3.37 $\pm$ 0.64 | 3.17 $\pm$ 0.51 | 2.66 $\pm$ 0.49 |

Table 3: **Performance of ProtMamba and other models at homolog-conditioned generation.** We report two structural scores, namely the pLDDT from ESMFold (Lin et al., 2023) and the scPerplexity from ProteinMPNN (Dauparas et al., 2022) for a set of 250 protein sequences generated using ProtMamba, each from a different cluster in our test set. Note that scPerplexity is the self-consistency Perplexity computed by ProteinMPNN from the ESMFold structures obtained for each generated sequence. We compare these values to those obtained for 250 protein sequences generated by EvoDiff-MSA, MSA-Transformer and Potts models, retrieved from the Zenodo archive associated to the EvoDiff paper (Alamdari et al., 2023), and which were generated each from a different cluster of the EvoDiff validation set. We also compare to a subset of the same size of natural sequences sampled from the same test set clusters as for ProtMamba.
$\uparrow$ (resp. $\downarrow$) indicates that higher (resp. lower) scores are better.

Finally, in Table 3, we compare the generative ability of ProtMamba to that of other models that can perform sequence generation conditioned on homologs from a specific protein family. Following the approach of Alamdari et al. (2023), we randomly sample 250 clusters from our test set, and, for

each cluster, we generate a sequence using ProtMamba conditioned on homologs randomly sampled from the cluster. We then compare two structural scores of these generated sequences with those obtained for sequences generated by EvoDiff-MSA (Alamdari et al., 2023), MSA Transformer (Rao et al., 2021) and Potts models (Russ et al., 2020) and provided in Alamdari et al. (2023), and for natural sequences. We find that ProtMamba outperforms existing models on the homolog-conditioned generation task, and generates sequences that obtain scores comparable to those of natural sequences. Note that the Hamming distance to the closest natural homolog of the sequences sampled using ProtMamba is $0.56 \pm 0.10$, similar to the value obtained for natural sequences ($0.48 \pm 0.17$) and not smaller, consistently with Figure 3.

## 4 DISCUSSION

Here, we presented ProtMamba, a homology-aware but alignment-free generative protein language model. ProtMamba leverages the long-context capabilities of state space models, allowing it to handle concatenated sequences of homologous proteins. It also benefits from their faster speed compared to attention-based models (Gu & Dao, 2023), allowing fast sequence generation and mutational effect prediction. ProtMamba was trained using a hybrid strategy combining autoregressive modeling and masked language modeling via the FIM objective. This allows ProtMamba to efficiently predict the next amino acid in a protein sequence as well as to inpaint masked regions.

Our results demonstrate ProtMamba's versatility across multiple tasks, including conditioned generation and protein fitness prediction, both for close and for distant variants. For homolog-conditioned generation, ProtMamba outperforms the state-of-the-art model EvoDiff-MSA (Alamdari et al., 2023). For fitness prediction, the sequence inpainting abilities of ProtMamba, via the FIM objective, proved to be particularly useful. Indeed, this functionality allows the model to exploit the full sequence context, without restricting to previous tokens as with autoregressive generation. This allows Prot-Mamba to reach similar performance levels as larger models, in a fraction of the time. Overall, ProtMamba benefits from capturing signal across multiple scales. In particular, it is able to predict fitness by exploiting constraints shared broadly across the proteome via its pre-training, but also specific constraints shared between homologs via the context, and it can exploit the full context of a given protein sequence when predicting only part of it.

**Limitations.** So far, ProtMamba did not reach perplexity values as low as those of larger transformer models like PoET (Truong Jr & Bepler, 2024) for full sequences. However, it can handle longer context sizes and requires much shorter training and inference times, which is extremely beneficial for the sequence inpainting task. We believe that scaling the model to larger sizes and training times (comparable to PoET) may result in comparable performance, while retaining ProtMamba's assets of lower memory cost and inference time.

We did not provide a direct test of the generative ability of ProtMamba for protein sequence inpainting. Indeed, this is a highly specific task lacking clear benchmarks so far. However, we believe that our two analyses on fitness prediction constitute a convincing indirect proof of the usefulness of ProtMamba's inpainting ability. It would be very interesting to experimentally test ProtMamba's inpainting ability, as well as its de novo sequence generation ability (Verkuil et al., 2022).

**Perspectives.** Our results demonstrate ProtMamba's flexibility, as it allows for precise conditioning by carefully choosing the context information (e.g. restricting to active sequences). Thus, ProtMamba responds very well to prompt engineering. We propose that this could become an alternative or complement to fine-tuning of language models. ProtMamba is also naturally designed to take advantage of retrieval augmented generation (RAG) techniques (Lewis et al., 2021), as it allows for using retrieved protein sequences from any external database, to condition the generation process.

Furthermore, we envision the possibility to use the model for homology search, by scoring sequences within specific contexts. This would be very fast, because only one forward pass would be required.

An interesting further extension of ProtMamba would be to make it explicitly structure-aware, e.g. using a structural alphabet (van Kempen et al., 2023), along the lines of SaProt (Su et al., 2023) or ProstT5 (Heinzinger et al., 2023). Another possible extension would be to include Gene Ontology (GO) terms to condition sequence generation (Madani et al., 2023; Nijkamp et al., 2023).

## 5 REPRODUCIBILITY STATEMENT

We describe in details all the steps to reproduce our work in Section 2 and we provide all the code in the supplementary material attached to the submission.

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

# Supplementary material

## A PROTEINGYM ASSAYS USED IN VALIDATION

Here, we list the 20 assays we extracted from the ProteinGym benchmark to choose some hyperparameters (see Figure S8).

A0A2Z5U3Z0_9INFA_Wu_2014
AMFR_HUMAN_Tsuboyama_2023_4G3O
CAR11_HUMAN_Meitlis_2020_lof
CBS_HUMAN_Sun_2020
CUE1_YEAST_Tsuboyama_2023_2MYX
DYR_ECOLI_Nguyen_2023
GDIA_HUMAN_Silverstein_2021
HIS7_YEAST_Pokusaeva_2019
HXK4_HUMAN_Gersing_2023_abundance
KCNE1_HUMAN_Muhammad_2023_expr

KKA2_KLEPN_Melnikov_2014
PITX2_HUMAN_Tsuboyama_2023_2L7M
PPM1D_HUMAN_Miller_2022
R1AB_SARS2_Flynn_2022
RDRP_I33A0_Li_2023
S22A1_HUMAN_Yee_2023_abundance
SCN5A_HUMAN_Glazer_2019
SHOC2_HUMAN_Kwon_2022
TRPC_SACS2_Chan_2017
VILI_CHICK_Tsuboyama_2023_1YU5

## B DETAILS ON MUTATIONAL EFFECT PREDICTION WITH PROTMAMBA

In Table 1, we consider different ways of computing mutational effects using ProtMamba, and compare with other models. Here, we explain our different approaches in more detail.

- The entry **ProtMamba (single)** of Table 1 is the result reported when using the FIM technique at the end of the sequence to evaluate mutations. It is the fastest method among homology-aware ones. The procedure is the following:
  1. Subsample a predetermined number of homologs of the target sequence considered in the DMS to be used as context, based on a diversity filter.
  2. Run ProtMamba on the context and collect the last hidden state of the context (as the model is recurrent).
  3. Start from the last hidden state as initial state for every variant to score. Scoring is then very fast, since we only need to apply ProtMamba on a single sequence per variant, and not on the full context. As Mamba scales linearly in sequence length, this allow to evaluate many different variants very fast on a single GPU (hundreds to thousands per batch). The mutated residues are put at the end of the sequence in the FIM mask. For single mutations, we can then evaluate in one shot the likelihood of every mutation.
  4. We compare the likelihood of the WT to the likelihood of the variant using Fill-in-the-middle to get a proxy of variant fitness. We can then evaluate the fitness of variants comprising each of the 20 amino-acids at the mutated site(s) in a single shot.

- The entry **ProtMamba AR (autoregressive)** of Table 1 is the result reported when evaluating the likelihood of a variant without using the FIM technique. The procedure is:
  1. Perform steps 1 and 2 as in ProtMamba (single) above.
  2. Start from the last hidden state as initial state for every variant to score. We then evaluate the autoregressive likelihood of the full variant. Since some logits are computed after the mutation in this setup (because they are positioned after in the sequence), we cannot evaluate the fitness of the 20 amino-acids in a single shot (in contrast to the previous approach), which increases the number of calls to ProtMamba, and hence requires more time.
  3. Compare the likelihood of the WT over the full sequence to the likelihood of the variant to get an evaluation of variant fitness.

- The entry **ProtMamba (w/ R), i.e. with retrieval** of Table 1 is the result reported when combining ProtMamba with a prior based on the frequency of amino-acids in the MSA of the relevant protein family.
  1. Perform steps 1, 2 and 3 as in ProtMamba (single) above.

2. Load the MSA, compute a log-likelihood prior ($\log p_{\text{retrieval}}$) based on the frequency of every amino-acid, and sum it with ProtMamba's log-likelihood ($\log p_{\text{ProtMamba}}$):

$$\log p_{\text{ProtMamba (w/R)}} = \alpha \log p_{\text{retrieval}} + (1 - \alpha) \log p_{\text{ProtMamba}} .$$

The parameter $\alpha$ was optimized using the validation set introduced in Section A (see Figure S8). This operation can be parallelized across CPUs or GPUs. We report the results using 16 workers.

Note that we tested ProtMamba AR both with our model fine-tuned on FIM (ProtMamba Long Finetuned) and with our foundation model (ProtMamba Long), and we obtained similar results (respectively 0.361 and 0.367).

In Table S1, we break down results by MSA depth and by number of mutations. We observe that for datasets with more than one mutation (last column in Table S1), ProtMamba with retrieval slightly outperforms the overall state-of-the-art model TranceptEVE L and reaches performance close to the structure-based model ESM-IF1. However, averaging over all datasets, ProtMamba does not reach the same performance as TranceptEVE L. But since ProtMamba performs better than Tranception L, ensembling ProtMamba and EVE predictions might yield comparable performance.

| Model | Par. | Spearman correlation by MSA depth | | | | by mutations | |
|---|---|---|---|---|---|---|---|
| | | All depths | Deep | Medium | Shallow | 1 | 2+ |
| ESM-2 | 150M | 0.387 | 0.497 | 0.358 | 0.306 | 0.367 | 0.379 |
| ESM-IF1 | 142M | 0.422 | 0.544 | 0.431 | 0.300 | 0.413 | 0.471 |
| Tranception S (w/o R) | 85M | 0.303 | 0.320 | 0.295 | 0.258 | 0.293 | 0.262 |
| Tranception L (w/o R) | 700M | 0.374 | 0.419 | 0.371 | 0.358 | 0.358 | 0.390 |
| **ProtMamba (w/o R)** | 107M | 0.406 | 0.465 | 0.411 | 0.391 | 0.376 | 0.444 |
| MSA Transformer | 100M | 0.421 | 0.473 | 0.435 | 0.393 | 0.392 | 0.435 |
| Tranception S (w/ R) | 85M | 0.418 | 0.444 | 0.415 | 0.428 | 0.389 | 0.409 |
| Tranception L (w/ R) | 700M | 0.434 | 0.473 | 0.438 | 0.432 | 0.404 | 0.463 |
| TranceptEVE L | >700M | 0.456 | 0.492 | 0.467 | 0.451 | 0.426 | 0.467 |
| **ProtMamba (w/ R)** | 107M | 0.432 | 0.472 | 0.438 | 0.448 | 0.404 | 0.469 |

Table S1: **Performance of different models on the ProteinGym benchmark.** We report Spearman correlation values obtained both based on retrieval (w/ R) and non-retrieval (w/o R) methods, and parameter count for each model. We report results divided according to MSA depth and number of mutations in the benchmark dataset. Results for benchmark models were obtained from `https://proteingym.org/`. Note that PoET-205M (Truong Jr & Bepler, 2024) reports an overall Spearman correlation of $0.474$ (Truong Jr & Bepler) on ProteinGym, but it is not yet on the ProteinGym website, and no information is given about the training time or resources.

## C  ABLATION STUDIES

We investigated ablations or alternative implementations of ProtMamba using two models: a small model with 14 million parameters (8 layers, hidden dimension 512) and the standard architecture with 107 million parameters (16 layers, hidden dimension 1024). Both models were trained for 10 billion tokens (50k steps with a batch size of 128) and evaluated on a validation set of 500 unseen clusters. During evaluation, a context of 25 sequences was used. The perplexity of the models was assessed both autoregressively (left-to-right) and in the fill-in-the-middle (FIM) spans.

The performance of these alternatives (described in more detail below) is summarized in Table S2. Perplexity values are reported for both autoregressive and FIM modes in the small model and the larger one.

| Perplexity | 14M Parameters | | 107M Parameters | |
|---|---|---|---|---|
| | **Autoregressive** | **FIM** | **Autoregressive** | **FIM** |
| Only FIM from scratch | Fail | $13.90 \pm 0.34$ | Fail | $15.59 \pm 0.27$ |
| AR only | $\mathbf{12.58} \pm 0.31$ | $18.03 \pm 0.25$ | $11.05 \pm 0.36$ | Fail |
| No positional encoding | $13.01 \pm 0.30$ | $16.71 \pm 0.47$ | $12.31 \pm 0.37$ | $17.20 \pm 0.58$ |
| Additive positional encoding | $12.72 \pm 0.31$ | $13.60 \pm 0.33$ | $12.58 \pm 0.38$ | $13.81 \pm 0.31$ |
| One mask, one token | $12.76 \pm 0.31$ | $15.54 \pm 0.29$ | $11.04 \pm 0.33$ | $16.60 \pm 0.36$ |
| Masking fraction 50% | $13.02 \pm 0.31$ | $\mathbf{13.44} \pm 0.33$ | $\mathbf{10.94} \pm 0.36$ | $\mathbf{11.59} \pm 0.35$ |
| ProtMamba | $13.00 \pm 0.30$ | $13.89 \pm 0.32$ | $11.35 \pm 0.33$ | $12.62 \pm 0.30$ |

Table S2: **Alternatives to ProtMamba.** Perplexity values are reported for different alternatives to ProtMamba, evaluated on small (14M parameters) and larger (107M parameters) models, for both autoregressive and FIM tasks.

The alternative implementations tested against our main ProtMamba model, and whose performance is reported in Table S2, were constructed as follows:

- **Only FIM from scratch:** This approach backpropagates the loss exclusively from the FIM tokens, disregarding the main amino-acid chain. Training this way from scratch disables autoregressive (left-to-right) next-token prediction and degrades performance, including on FIM tasks.

- **Autoregressive (AR) only:** Trains the model without sampling FIM spans. While this slightly improves autoregressive performance, it significantly degrades FIM capabilities.

- **No positional encoding:** Omits positional encodings entirely. In autoregressive mode, the model can partially rely on its recurrent architecture, but in FIM mode, performance suffers due to the absence of positional information in input.

- **Additive positional encoding:** Uses additive positional encoding (summing token embeddings with positional encoding) instead of concatenated positional encoding (concatenating token embeddings with positional encoding). This approach showed mixed results, with slight improvement in the small model but degradation in the larger model.

- **One mask, one token:** Uses one mask per token (as in the T5 model (Raffel et al., 2020)) instead of one mask per span of tokens (as in our approach, inspired by Bavarian et al. (2022)). This approach led to performance degradation in FIM, likely due to insufficient training on larger number of mask tokens.

- **Masking fraction 50%:** Samples 50% of the tokens for FIM (compared to ProtMamba's 20%). This alternative brought minor but noticeable improvements, suggesting potential for further development.

## D   SUPPLEMENTARY FIGURES

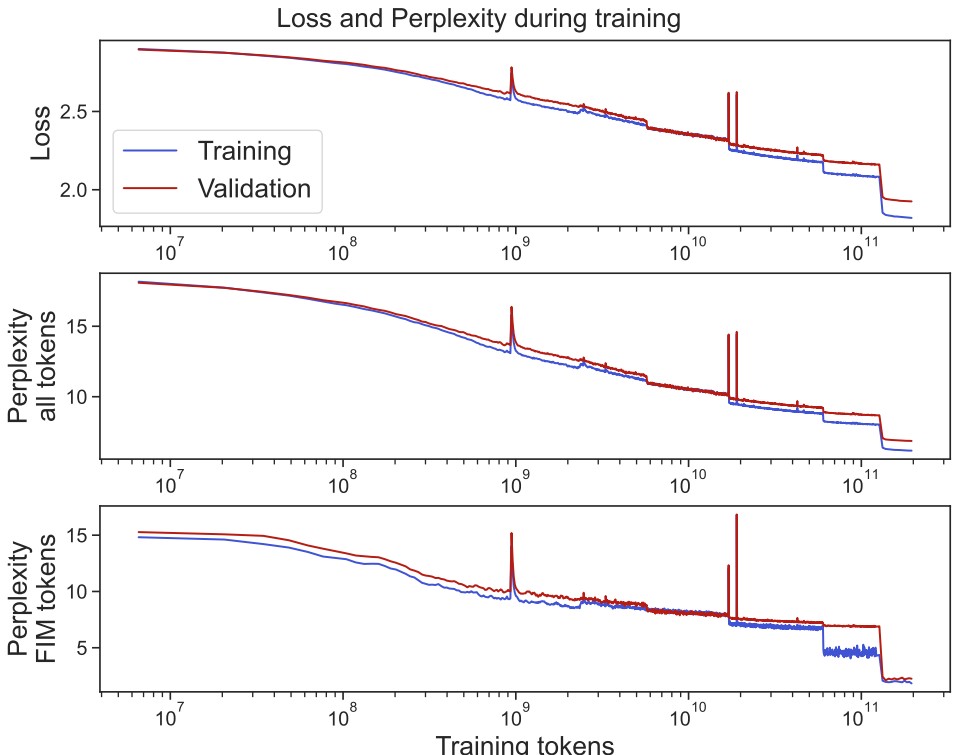

Figure S1: **Loss and perplexity during training.** Cross entropy loss and perplexity computed for both the full non-masked sequences and the FIM tokens. We show them as a function of the number of tokens processed during the training of ProtMamba. They are computed on the training set and on a validation set of 192 held-out OpenProteinSet sequence clusters (see Section 2.3).

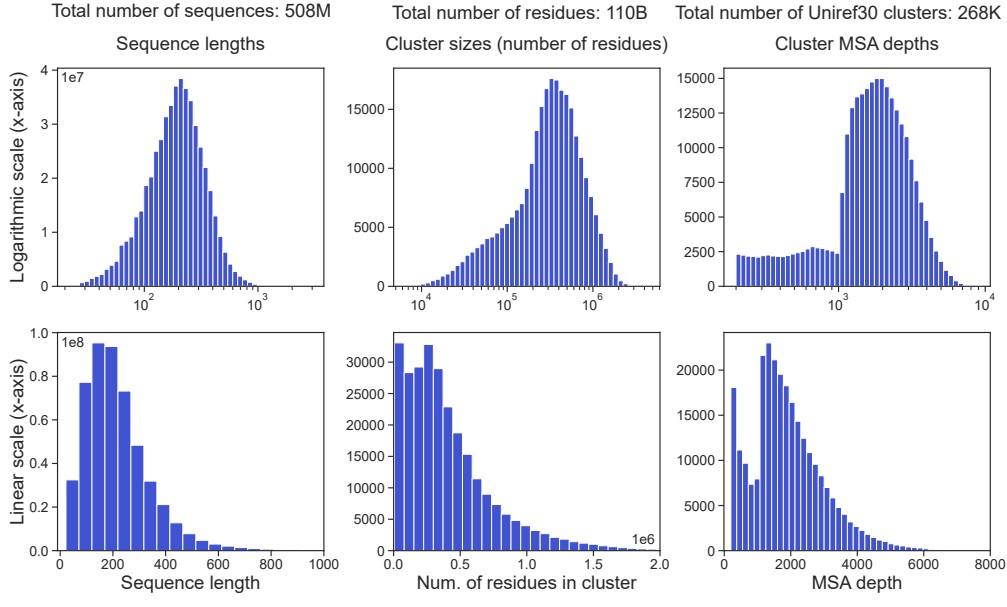

Figure S2: **Datasets statistics.** We show the x axis both in log scale (first row) and in linear scale (second row) to have a better grasp of the distributions.

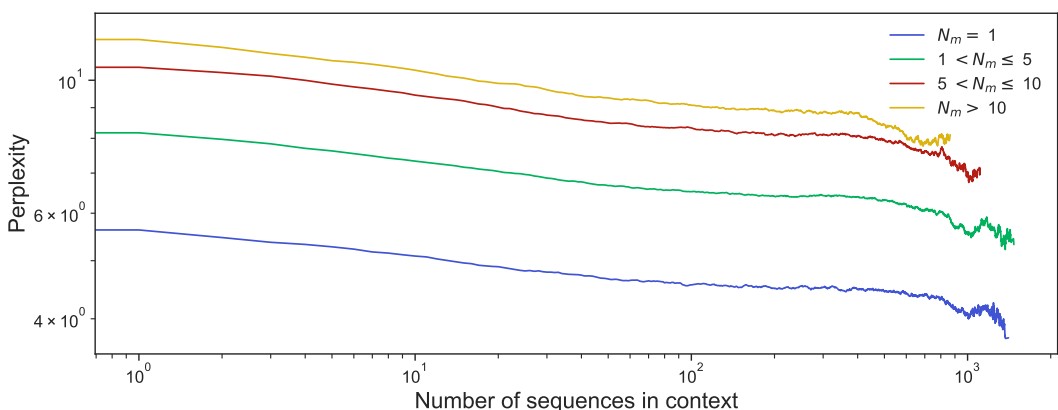

Figure S3: **Scaling of the FIM perplexity with the number of context sequences.** Same as Figure 2, using logarithmic scales on both axes.

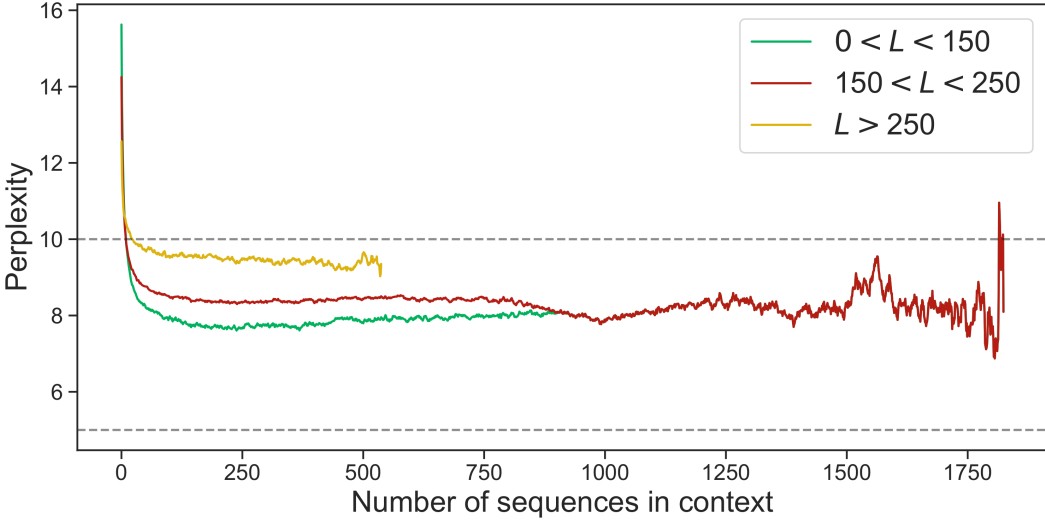

Figure S4: **Loss and perplexity of the full sequences vs. number of sequences in the context.** Scaling of the per-sequence perplexity (i.e. the standard autoregressive perplexity of the full non-masked sequence) versus the number of context sequences. Results are averaged over all 500 clusters of the test set and 20 replicates for each cluster (differing by the random sampling of context sequences). Context sizes go up to $2^{17}$ amino acids. Sequence clusters are split according to the average length $L$ of sequences in the cluster. We observe that clusters with shorter sequences reach lower perplexities.

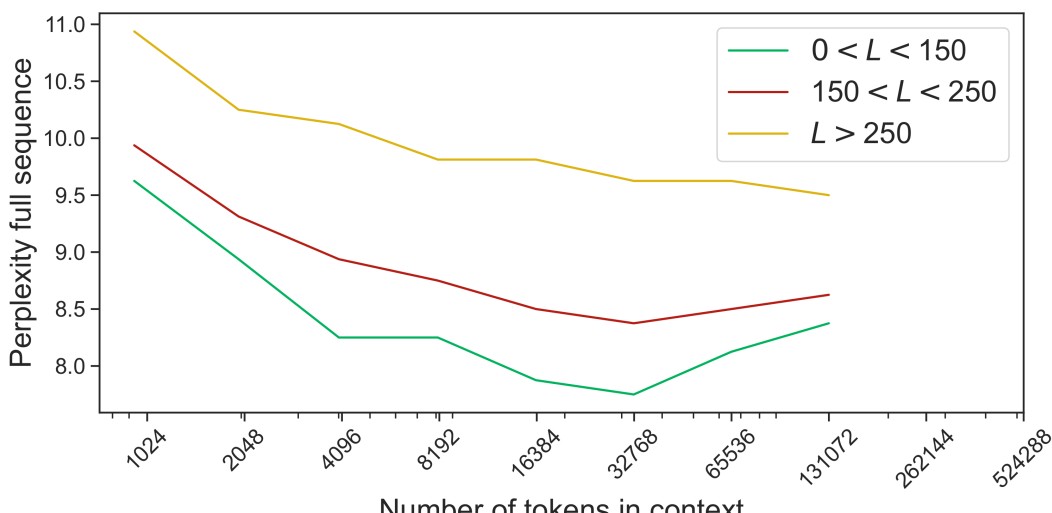

Figure S5: **Loss and perplexity of the full sequences vs. number of tokens in the context.** Scaling of the per-sequence perplexity (i.e. the standard autoregressive perplexity of the full non-masked sequence) versus the size of the context (i.e. the number of preceeding tokens). Results are averaged over all 500 clusters of the test set and 20 replicates for each cluster (differing by the random sampling of context sequences). Context sizes go up to $2^{17}$ amino acids. Sequence clusters are split according to the average length $L$ of sequences in the cluster. We observe that clusters with shorter sequences reach lower perplexities.

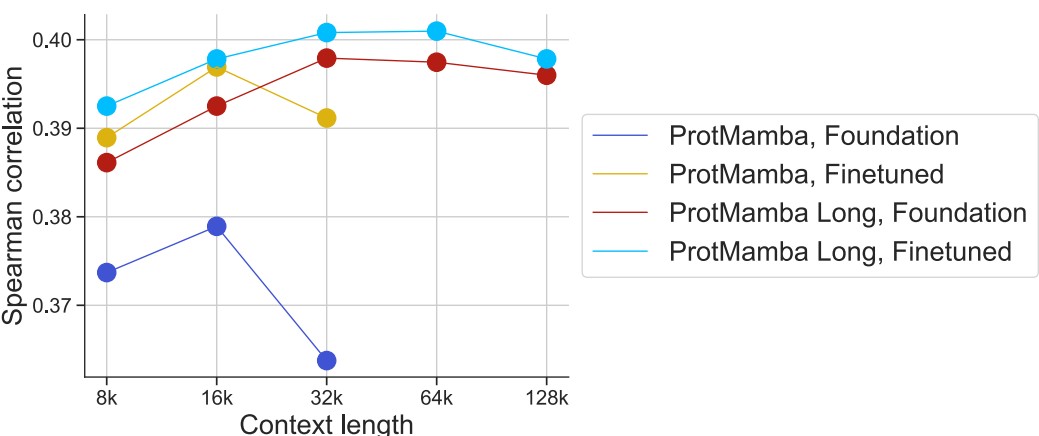

Figure S6: **Comparison of 4 ProtMamba variants on the ProteinGym benchmark.** We show the predictive power for variant effect on the ProteinGym benchmark, via the Spearman correlation between predictions and experimental results, for "ProtMamba, Foundation" ($2^{15} = 32768$ tokens context seen in training), "ProtMamba, Fine-tuned" (fine-tuned on predicting only FIM tokens), ProtMamba Long, Foundation" ($2^{17} = 131072$ tokens context seen in second phase of training) and "ProtMamba Long, Fine-tuned" (fine-tuned on predicting only FIM tokens). We notice that models fine-tuned only on the FIM objective outperform the foundation models. ProtMamba Long is overall performing better than ProtMamba and its performance does not decrease as sharply as ProtMamba for longer context.

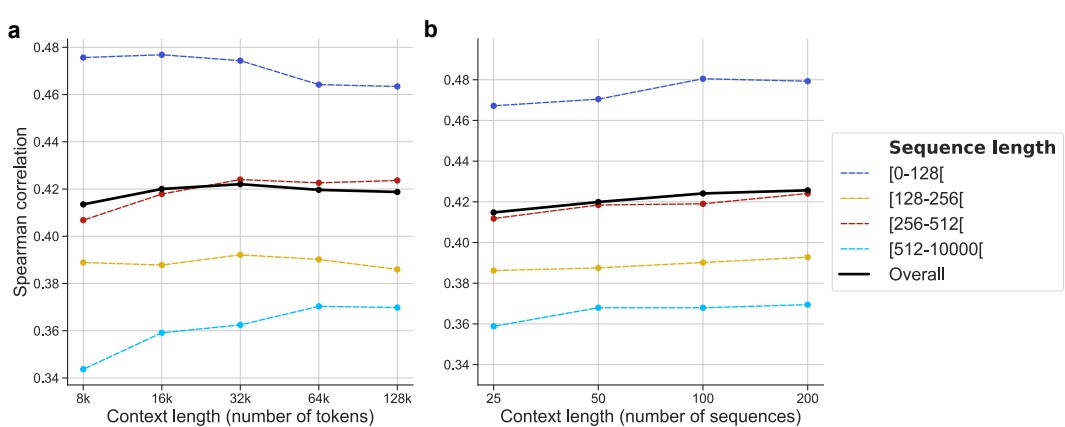

Figure S7: **Impact of context length on results on the ProteinGym benchmark.** **(a)** We run ProtMamba Long on the ProteinGym dataset, building contexts of different sizes in terms of numbers of tokens (from 8,000 to 128,000). We see that the increase in performance is more important for long sequences, which highlights the benefit of long context to model long protein sequences. **(b)** We also run ProtMamba Long on the ProteinGym dataset, building contexts of different sizes in terms of numbers of sequences (from 25 to 200). Overall, we notice a rise in the Spearman correlation, showing that prediction benefits from longer context.

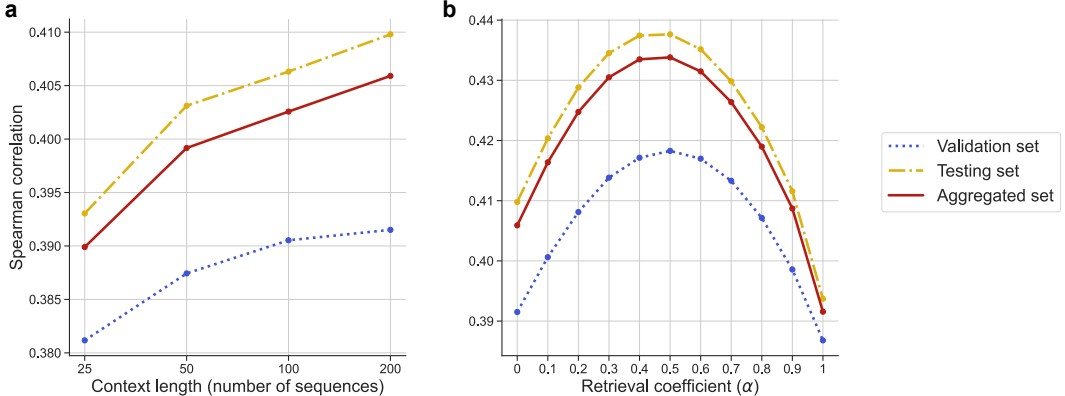

Figure S8: **Choice of context length and retrieval coefficient using a validation set.** We randomly extracted a validation set of 20 datasets (see supplementary Section A) to select the best context length and retrieval coefficient. **(a)** The prediction improves with the context size in the validation set. This trend was later observed in the rest of the benchmark (testing set) too. **(b)** Retrieval requires mixing the fitness score $\mathcal{F}_m$ obtained from ProtMamba and the fitness score obtained from the independent-site model $\mathcal{F}_i$ through the retrieval fitness score $\mathcal{F}_r = \alpha\mathcal{F}_i + (1 - \alpha)\mathcal{F}_m$. The best model on the validation set was obtained for a retrieval coefficient $\alpha = 0.5$, which was later verified on the rest of the dataset.

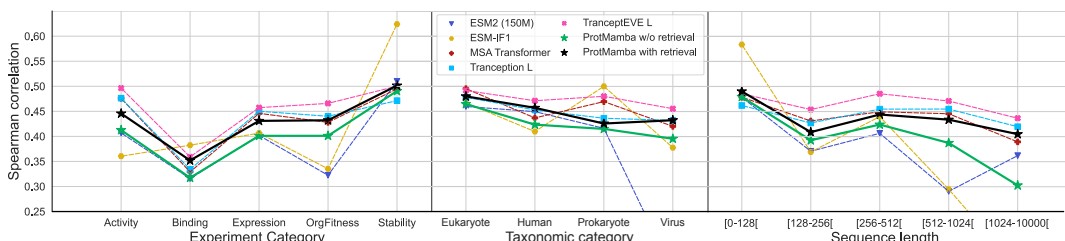

Figure S9: **Breakdown by categories of results on the ProteinGym benchmark.** Results of ProtMamba Long and of existing specialized models on the ProteinGym benchmark, averaged over all datasets, are shown broken down by category of experiments (left), taxonomic category (middle) and wild-type sequence length (right). ProtMamba is fairly competitive with these models. We note that the inverse folding model ESM-IF1 outperforms sequence-based models for stability assessment, as expected (see left panel).

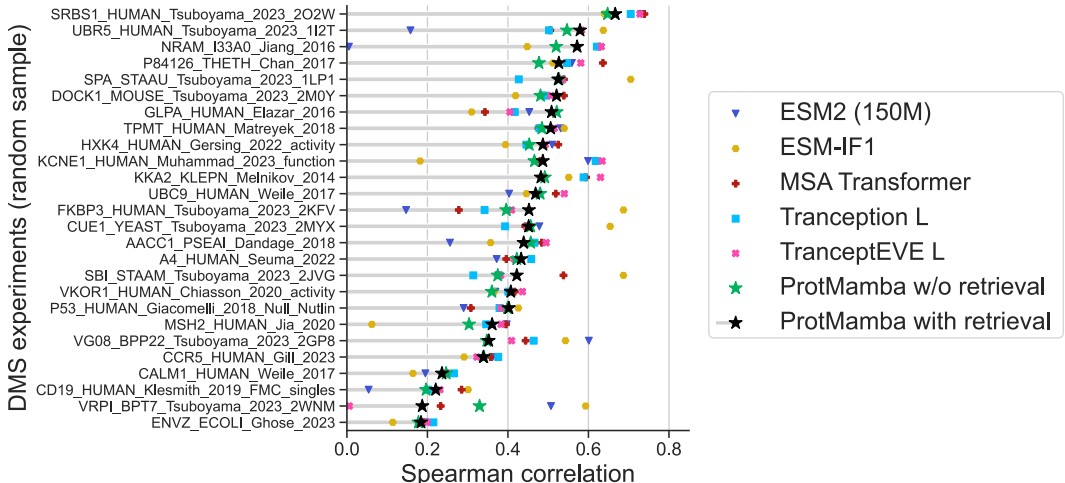

Figure S10: **Example results on the ProteinGym benchmark.** Results of ProtMamba Long are shown on 25 randomly sampled deep mutational scan (DMS) experimental datasets from ProteinGym, and are compared to existing methods (see main text). The score shown is the Spearman correlation between predictions and experimental results.

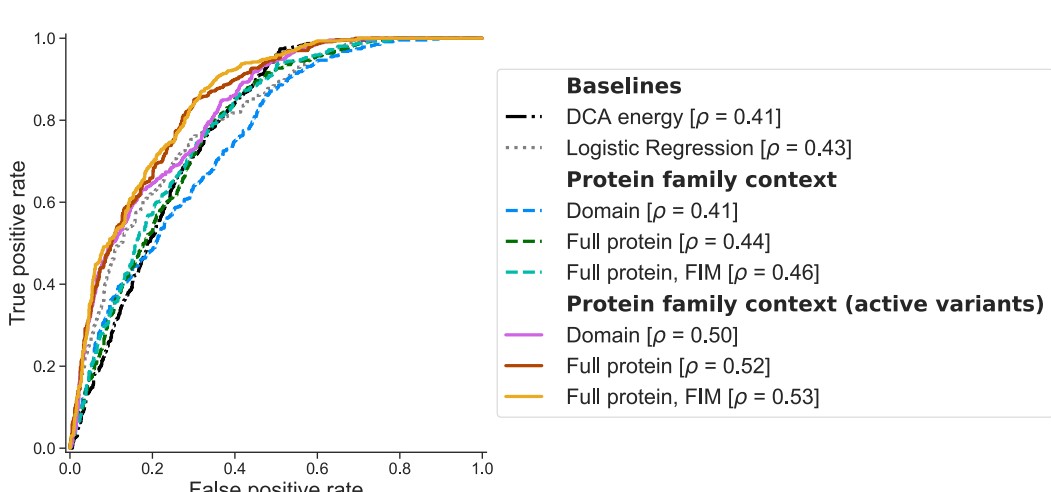

Figure S11: **Impact of various context construction methods on results on chorismate mutase activity.** The ROC curve is shown for various context construction methods (see main text) for predicting active variants in the chorismate mutase dataset, and for baseline methods from Russ et al. (2020). Overall, we observe that restricting to active variants in context helps improving prediction quality (Spearman correlation $\rho$ going from 0.41-0.46 to 0.50-0.53). Giving full proteins instead of restricting to the chorismate mutase domain also improves the results. Using FIM to condition the domain to score using the rest of the protein also improves performance. ProtMamba also outperforms the baselines provided in Russ et al. (2020), namely the Potts or DCA energy and the logistic regression trained directly on amino-acid sequences.

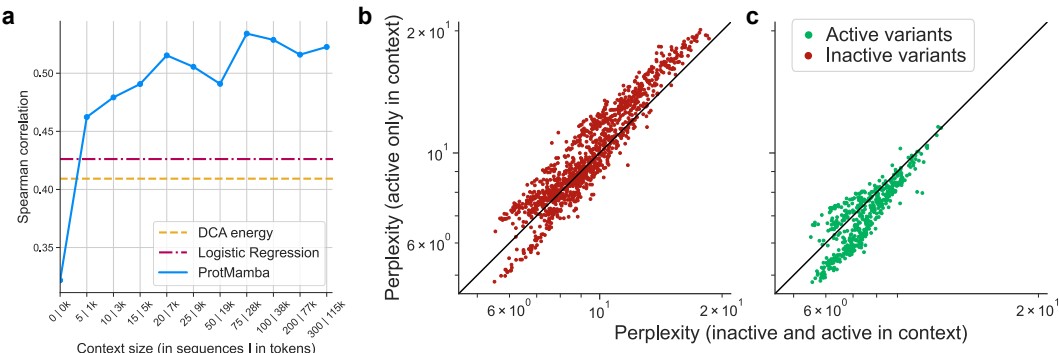

Figure S12: (**a**) **Impact of context length on results on chorismate mutase activity.** Spearman correlation between experimental activity and predictions from ProtMamba is shown using a different number of active sequences in the context (using FIM and full active proteins as context to score variants using ProtMamba). The Spearman correlation quickly increases with the number of proteins sequences given in context, especially from 0 to 25 sequences (or 10,000 tokens) before slowly increasing with context size. (**b**) and (**c**) **Perplexity of generated variants when using only active variants in context (b) or using active and inactive variants in context (c)**. Inactive variants tend to have higher perplexity (implying lower fitness score) when the context contains only active variants (**b**) while active variants have lower perplexity (implying higher fitness score) when the context contains only active variants (**c**) .

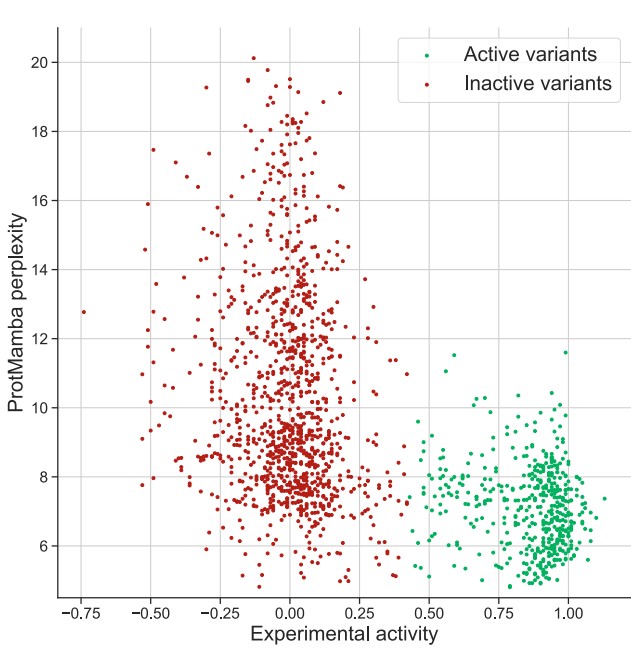

Figure S13: **ProtMamba captures chorismate mutase activity.** Experimental activity of chorismate mutase enzyme variants from Russ et al. (2020) is shown versus ProtMamba per-token perplexity, determined using FIM and full active proteins as context. The per-token perplexity is a good proxy of the activity. We obtain a Spearman correlation of 0.53 between this score and experimental activity, and it yields an AUC of 0.84 to discriminate active from inactive sequences.

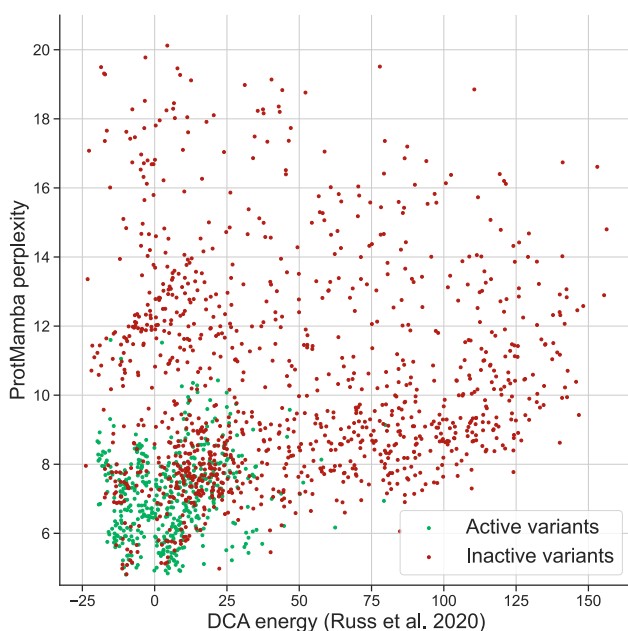

Figure S14: **ProtMamba perplexity versus DCA energy for chorismate mutase variants.** Prot-Mamba perplexity is evaluated using full sequences, FIM and only active variants in the context, and is shown versus the Potts or DCA energy from Russ et al. (2020). Active variants are in green, while inactive variants are in red. We observe that most of the variants that are active have low perplexity, and that many inactive variants that were not discriminated as inactive by DCA are labelled as such by ProtMamba (bottom right part of the plot).

| Cluster | Hamming | HMMER | pLDDT | pTM |
|---|---|---|---|---|
| A0A2H9MP70 | 0.45 | −0.44 | −0.77 | −0.54 |
| A0A135YUE9 | 0.76 | −0.68 | −0.61 | −0.58 |
| G4ZH78 | 0.23 | −0.42 | −0.47 | −0.47 |
| A0A1A8YWK1 | 0.79 | −0.7 | −0.84 | −0.81 |
| A0A0A0HZM8 | 0.46 | −0.47 | −0.69 | −0.63 |
| A0A091TDH7 | 0.31 | −0.32 | −0.46 | −0.46 |
| A0A2N1P554 | 0.18 | −0.35 | −0.37 | −0.4 |
| A0A1C5UJ41 | 0.81 | −0.78 | −0.81 | −0.77 |
| A0A194V424 | 0.66 | −0.71 | −0.7 | −0.54 |
| S7UZ45 | 0.56 | −0.57 | −0.74 | −0.66 |
| F2CV06 | 0.89 | −0.85 | −0.81 | −0.79 |
| A0A146ZGL6 | 0.72 | −0.75 | −0.56 | −0.64 |
| D8SD16 | 0.65 | −0.74 | −0.71 | −0.61 |
| A0A139IN77 | 0.18 | −0.2 | −0.15 | −0.27 |
| A0A1C6Q5J2 | 0.55 | −0.22 | −0.45 | −0.27 |
| I4B642 | 0.44 | −0.52 | −0.61 | −0.52 |
| A0A2X4BAY2 | 0.27 | −0.63 | −0.63 | −0.54 |
| A0A241VGM5 | 0.44 | −0.59 | −0.62 | −0.55 |
| A0A1S3G530 | 0.88 | −0.72 | −0.8 | −0.74 |
| **Mean** | 0.54 | −0.56 | −0.62 | −0.57 |

Figure S15: **Pearson correlation between ProtMamba perplexity and scores for generated sequences.** For each of 19 test clusters, we used all the sequences generated by ProtMamba to compute the Pearson correlation between the model perplexity and the Hamming distance to the closest natural neighbor, the HMMER score, the pLDDT and pTM scores from ESMFold.

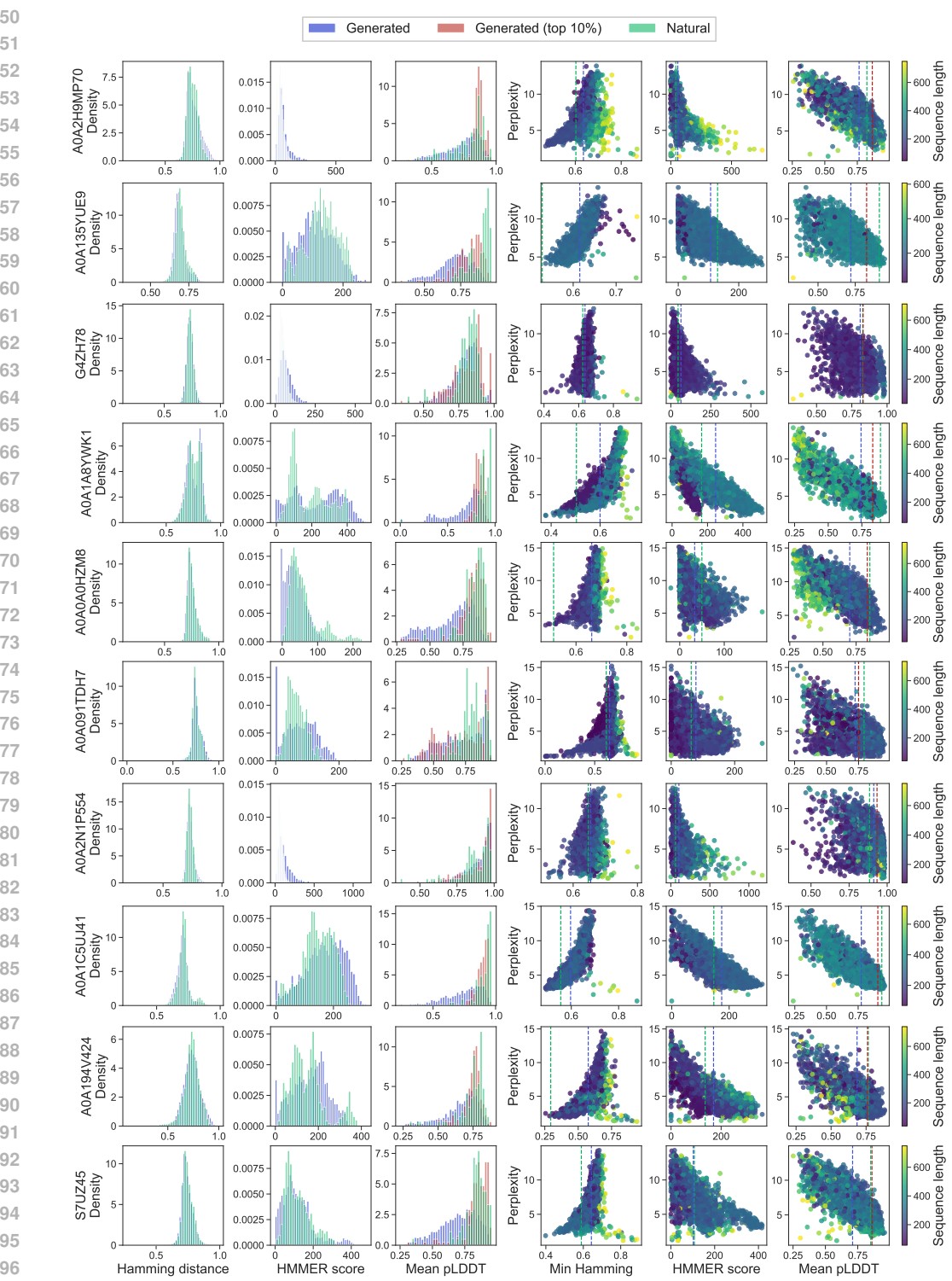

Figure S16: **Properties of generated sequences.** Left panels: histograms of Hamming distances, HMMER scores and mean pLDDT scores from ESMFold of generated sequences for 10 example test clusters (10 rows). Right panels: scatter plots of ProtMamba perplexity versus the Hamming distance to the closest natural neighbor, the HMMER score and mean pLDDT score from ESMFold for all generated sequences from each of 10 example clusters (10 rows). Dashed vertical lines: median of the generated sequences (blue), median of the natural sequences (green) and pLDDT value of the reference structure of the cluster (red). The last one is shown only for the rightmost plot.

