# OpenReview forum: "ProtMamba: a homology-aware but alignment-free protein state space model"
_ICLR.cc/2025/Conference — Submitted to ICLR 2025_

### Official Review · Reviewer_Ua6N · 2024-10-21

**Soundness:** 3
**Presentation:** 3
**Contribution:** 3
**Rating:** 6
**Confidence:** 3

**Summary:**

The paper presents ProtMamba, a homology-aware but alignment-free protein language model. It's based on the Mamba architecture and trained on concatenated homologous sequences. Results show its effectiveness in various tasks like sequence generation and fitness prediction.

**Strengths:**

- The authors propose a new training strategy, which effectively harnesses evolutionary information from homologous sequences without relying on MSA.
- The architecture based on Mamba blocks allows for handling extremely long contexts, which is beneficial for protein modeling as concatenating homologous sequences often results in long inputs.
- The results are comprehensive and prove the effectiveness of the proposed model.

**Weaknesses:**

I am not certain whether combining protein language with mamba can be regarded as "novel", but it is ok for me since such combination is not explored yet. An interesting aspect of this study is the training paradigm, which might provide insights for future studies. Nevertheless, a disappointing point is that almost no ablation study of the method can be found.

- The mask strategy (span-mask) is similar to that of T5 (you'd better add the missing reference: Raffel et al, Exploring the Limits of Transfer Learning with a Unified Text-to-Text Transformer). There are some most related strategies that are not ablated, such as:
  - what if training with token-level mask instead of span-mask? Token-level mask means something like: "a b c <m1> <m2> <m3> g h  <eos> <m1> d <m2> e <m3> f".
  - what if no masking strategy is used and the model is trained in an autoregressive fashion?
  - what if doing mask-prediction without observing subsequent tokens, which means the input is "a b c <m1> <m2> <m3> g h" while the target is "b c d e f g h <eos>"?
  - I am not requesting the authors to ablate all of the above. However, for an AI conference, that would be very interesting and no ablation is unacceptable (in my opinion).
- The authors claim that the incorporation of position embedding and the concatenation strategy are important. I think the authors can present some results for comparison, for example, a model without position embedding and a model with the addition strategy (of the position embedding).
- Absence of comparison with a transformer (with FlashAttention) in the same setting.
- Have you tried to scale up the parameter of the architecture?

**Questions:**

NA

---

> ### Author Response · Authors · 2024-11-19
> **Response (1/1)**
>
> We thank the reviewer for their valuable feedback on our paper. We performed multiple novel ablations and comparisons that we report in the main comment of the rebuttal. We also address the reviewer’s specific comments and concerns below:
> 1. **Span mask strategy:** We referenced the two previous papers from which we built our span mask strategy. We also cited the relevant reference suggested by the reviewer in the new version of the paper.
> 2. **Ablations on the masking strategy:** We performed the requested ablations (together with several others) and we reported their results in the common answer to all reviewers. In particular, we performed an ablation using a fully autoregressive model, and showed its performance compared to the others. Beyond these ablations, a purely autoregressive model, even if interesting, does not fulfill all our objectives. Indeed, the reason why we use FIM is that this allows us to perform inpainting of protein sequences, which would not be possible using a purely autoregressive model. Finally, we believe that the idea of having a mask size of 1 is quite interesting, and we performed some ablations showing that it is also good in a FIM objective. We leave exploring this direction in more detail to future works.
> 3. **More conclusive results on protein generation:** We would like to highlight that, based on other reviewers’ comments, we added a comparison with other models on the conditional protein generation task (discussed in the general rebuttal) that may be of interest for the reviewer.
> 4. **Positional encoding:** We performed ablations on the positional encoding and reported the results in the common answer to all reviewers. These ablations show that there is not much difference between the different ways of using positional encoding and not using positional encoding. The reason why we decided to employ positional encodings is that this allows us to control the length of the patches generated using the FIM objective during inference. Concretely, as shown in Fig 1 of the paper, by artificially skipping from position N and N+3 (after the mask token) in the main sequence, we are able to give the model the instruction of generating 3 FIM tokens as output. This would not be possible if we used no positional encoding. To summarize, positional encodings do not deteriorate performance and allow us to perform inpainting with length control.
> 5. **Comparison with a transformer model:** The reason why we did not benchmark ProtMamba against a vanilla transformer is that a transformer is not able to handle such a long context during training. Even a small transformer model with a much more limited context length would require at least an order of magnitude more compute and training time than a Mamba model. For this reason, we decided to rather provide a comparison with PoET, which is a published transformer model that takes homology into account.
> 6. **Model scaling:** We did not scale up the parameter count of the model yet because of hardware limitations. We believe that this work is important as a proof of concept, and we plan to train a larger model in a subsequent work.

---

> > ### Comment · Reviewer_Ua6N · 2024-11-25
> >
> > I have read the response and appreciate the authors' efforts to conduct further experiments and revise the manuscript. I would like to increase my score.
> >
> > It should be noted that a score of 6 results from the training strategy, rather than adopting an Mamba architecture. I believe that the training strategy is orthogonal to the Mamba framework. I am of the opinion that a Transformer can perform the same task (even better), considering that there are numerous large language models (LLMs) capable of handling long sequences (128k), and their size is larger than that of the model in this study. I strongly suggest that the authors add a Transformer architecture for comparison.
> >
> > Additionally, increasing the model size should also be an important future work.
> >
> > Thank you.

---

> > > ### Author Response · Authors · 2024-11-25
> > >
> > > We thank the reviewer for raising their score, and for their constructive feedback that allowed us to significantly improve our paper with respect to the initial version.
> > >
> > > Our choice of Mamba over Transformer-based architectures is primarily motivated by computational efficiency. While Transformers with FlashAttention can manage longer sequences with a linear memory-complexity, their inherent quadratic time-complexity in sequence length still imposes significant computational overhead during training and inference. In contrast, Mamba’s linear time complexity enables it to handle long sequences with far greater efficiency, making it more practical for scenarios where computational resources or time are constrained. Concretely, we show that we are able to obtain a 85 to 1300-fold speedup with respect to PoET, a transformer model trained on a similar objective. We believe that building efficient models can lead to their broader use in the community, and mitigate their carbon footprint.
> > >
> > > To address the reviewer’s concerns, we started training some experiments to compare ProtMamba with a small GPT2 transformer (with Flash-Attention) matching the size of the small Mamba model used in our ablations. The results of these experiments will be included in the final version of the paper and as a comment as soon as possible.
> > >
> > > We fully agree with the reviewer that increasing model size is an important direction for future work. Scaling ProtMamba would likely improve its performance across tasks, making it more competitive with larger language models that can handle extended sequences. In line with this, we also plan to explore the recently introduced Mamba2 architecture, which incorporates attention layers and has demonstrated improved performance over Mamba1.

---

> > > > ### Author Response · Authors · 2024-11-29
> > > > **Additional comparison with Transformer and Hyena baselines.**
> > > >
> > > > Thank you again for your valuable suggestions. Below, we provide additional results about comparisons between other models (which was requested by reviewer **biCn** too). We will report these results in the supplementary material in later revisions.
> > > >
> > > > - **Training Setup:** We conducted a comparative evaluation of Mamba, Hyena, and GPT2 using flash attention, focusing on small-scale models, with our training framework and data. Each model was designed to have approximately 14M parameters, consisting of 8 layers with 512-dimensional representations (384 for GPT2). All models were trained with a context size of 32,000 tokens for 72 hours on single H100 GPUs. Note that we also started training a larger GPT2 model with 115M parameters. However, during the first 24 hours, the loss was decreasing much more slowly than with Mamba and Hyena, so we decided to focus on small models instead (the time necessary to converge would have been too high for the timeframe of this rebuttal).
> > > >
> > > > - **Evaluation:** We tested the perplexity of the models on a held-out validation set with two context lengths: 16,000 tokens (short context) and 128,000 tokens (long context).
> > > > 1. For short contexts, we report the evaluation rate (number of sequences evaluated per second) and validation perplexity (PPL) to demonstrate efficiency under standard settings.
> > > > 2. For long contexts, we report the evaluation rate and validation perplexity for GPT2 and Mamba. Hyena, however, could not be evaluated with context lengths beyond 32,000 tokens, as it is limited to context lengths defined during training.
> > > >
> > > > - **Comments on the comparison:** The table below shows that Hyena has a much higher perplexity than the other two architectures. As expected, GPT2 is better at modeling short contexts than Mamba but its performance decreases with longer contexts (not seen in training). Moreover, Mamba reaches a performance similar to that of GPT2 at longer contexts. We also show that GPT2 is 16- to 128-fold slower than Mamba during evaluation, which can be limiting for many applications (e.g. variant effect prediction, see the comparison with PoET in Table 1 of the paper). Furthermore, the small GPT2 model has an inference time which is still 4- to 34-fold larger than the larger ProtMamba model that we trained (107M).
> > > >
> > > > - **Future Directions with Transformer models:** The original approach (training protocol, dataset construction) presented in this paper is flexible and can be implemented with other architectures. In particular, using GPT2 seems indeed a promising direction for improving performance. However, this comes with substantially higher computational costs (in inference time in particular), particularly for longer contexts, where GPT2 is less efficient than Mamba. Another promising direction involves building on the recent emergence of hybrid models that mix state-space models and attention-based methods.
> > > >
> > > > | Model         	| Training Time | Tokens Seen | Short Context PPL (↓) | Eval Rate: Short Context (↑)  | Long Context PPL (↓) |Eval Rate: Long Context (↑) |
> > > > |--------------------|---------------|-------------|--------------------|--------------------------|-------------------|--------------------------|
> > > > | Hyena (14M)   	| 72h       	| 81.9B    	| 14.42         	| 17 $s^{-1}$    	| Fail          	| Fail                	|
> > > > | Mamba (14M)   	| 72h       	| 76.8B    	| 12.48         	| 33 $s^{-1}$  	| 11.89         	| 3.84 $s^{-1}$	|
> > > > | GPT2 (15M)    	| 72h       	| 33.3B    	| 10.68         	| 2 $s^{-1}$    	| 11.72         	| 0.03 $s^{-1}$ 	|
> > > > | |
> > > > | ProtMamba (107M)  | 400h      	| 190B     	| 9.69          	| 8.33 $s^{-1}$  	| 9.26          	| 1.04 $s^{-1}$ 	|

---

### Official Review · Reviewer_biCn · 2024-10-27

**Soundness:** 3
**Presentation:** 2
**Contribution:** 2
**Rating:** 5
**Confidence:** 4

**Summary:**

The paper introduces **ProtMamba**, a novel protein language model that is homology-aware but alignment-free, addressing the limitations of traditional multiple sequence alignments (MSAs) in protein modeling. ProtMamba is built on the Mamba architecture, which enables it to handle very long sequences by efficiently processing concatenated homologous protein sequences. The model is trained using a hybrid of autoregressive modeling and Fill-in-the-Middle (FIM) objectives, making it highly versatile for tasks like protein sequence generation and mutational fitness prediction. ProtMamba demonstrates competitive performance on benchmarks like ProteinGym, outperforming similar-sized models in terms of efficiency and predictive accuracy. Additionally, the model excels in sequence generation tasks, producing novel sequences with structural properties comparable to natural proteins.

**Strengths:**

- ProtMamba is Homology-aware yet alignment-free
- Mamba architecture adaptation efficiently handle long contexts
- Explores hybrid training scheme for pLMs
- ProtMamba strong results on ProteinGym for mutational fitness prediction
- ProtMamba can generate reasonable sequences given homology context/sequences

**Weaknesses:**

1. Why use position encoding for ProtMamba? Given the recurrent nature of Mamba, positional information should theoretically be learned implicitly, which is why the original Mamba model does not employ explicit position encodings. The authors claim this is a significant modification, yet they fail to provide any experiments or ablation studies to demonstrate how this change improves model performance. I would like to see a comparison of with positional encoding (PE) vs. without PE, and additionally, a comparison of standard PE (additive) vs. the concatenation method used in ProtMamba. Authors should provide more details about PE implementation and comparison.

2. Although the motivation for using a long-context language model like Mamba is compelling, the paper does not benchmark a vanilla transformer at any context length, which is a significant weakness. Without this comparison, it is difficult to argue whether ProtMamba is the optimal architecture for this task in terms of performance. While it is clear that state space models (SSMs) like Mamba will likely outperform transformers in terms of efficiency, the lack of performance benchmarks makes it hard to assess if ProtMamba achieves the best results.

3. Regarding the ProteinGym benchmark, when incorporating MSA or homology sequences, ProtMamba does not outperform MSA-based models. This challenges the fundamental premise of the paper that a homology-aware, MSA-free model should perform better and eliminate the need for MSAs. The lack of superior performance compared to MSA-based models suggests that ProtMamba's approach may not fully leverage homology information as effectively as MSA based models.

**Questions:**

1. How do the authors compute the total number of tokens and FLOPs for training? Can you provide more details on the implementation, such as whether you use any packages for FLOP calculations or how you approximate them?

2. Why did you choose to use a Poisson distribution for masking instead of other distributions?

3. In Figure 12, why does the model, in some DMS experiments, perform better without retrieval (MSA/homology sequences) than with MSA? This is especially surprising given that the model was trained over long contexts of homologous sequences. For instance, in VRPI_BPT7_Tsuboyama_2023_2WNM, there is a significant difference. Could you provide some explanations for this discrepancy?

---

> ### Author Response · Authors · 2024-11-19
> **Response (1/1)**
>
> We thank the reviewer for their valuable feedback on our paper. We performed multiple novel ablations and comparisons that we report in the main comment of the rebuttal. We also address the reviewer’s specific comments and concerns below:
> 1. **Positional encoding:** We performed ablations on the positional encoding and reported the results in the common answer to all reviewers. These ablations show that there is not much difference between the different ways of using positional encoding and not using positional encoding. The reason why we decided to employ positional encodings is that this allows us to control the length of the patches generated using the FIM objective during inference. Concretely, as shown in Fig 1 of the paper, by artificially skipping from position N and N+3 (after the mask token) in the main sequence, we are able to give the model the instruction of generating 3 FIM tokens as output. This would not be possible if we used no positional encoding. To summarize, positional encodings do not deteriorate performance and allow us to perform inpainting with length control.
> 2. **Benchmarking against a transformer:** The reason why we did not benchmark ProtMamba against a vanilla transformer is that a transformer is not able to handle such a long context during training. Even a small transformer model with a much more limited context length would require at least an order of magnitude more compute and training time than a Mamba model. For this reason, we decided to rather provide a comparison with PoET, which is a published transformer model that takes homology into account.
> 3. **ProteinGym benchmark:** We agree with the reviewer that MSA information is still very important for performance on the ProteinGym benchmark. Accordingly, in our case, ProtMamba with retrieval improves over ProtMamba. We also show that ProtMamba without retrieval performs nearly as good as MSA-based methods like MSA-Transformer, with a fraction of their compute, and without using MSAs, which are not always available / not always have a good quality (as shown in point 6 of our answer). In addition, we believe that the usefulness of ProtMamba goes beyond this benchmark. Indeed,  ProtMamba is a generative model and can use homology information to conditionally generate novel sequences, while the MSA-based models in Table 1 cannot do so. Furthermore, in our common answer to reviewers, we reported the performance of ProtMamba on the conditional generation task, showing that it outperforms the MSA-based state of the art method EvoDiff. This confirms the interest of an MSA-free model like ProtMamba.
> 4. **Estimation of compute:** We estimated the compute by using the formula:
> Compute = (Training time) × (# of GPUs) × (Peak FLOP/s of GPU) × (Utilization rate, in our case ~80%).
> 5. **Poisson distribution for masking:** Our choice was based on the necessity of having a low number of masks, given that in the typical use-cases of the model (e.g. protein design), we expect that one will usually not want to mask more than a few regions (e.g. binding pocket) of the sequence to inpaint them. Besides, we do not expect this choice to have a strong impact on the training process.
> 6. **Performance on some specific ProteinGym datasets:** We further analyzed the performance of the model on the dataset pointed out by the reviewer, namely VRPI_BPT7_Tsuboyama_2023_2WNM, and on others. We noticed that in all ProteinGym datasets in which ProtMamba without retrieval outperforms ProtMamba with retrieval, all the other methods that are MSA-independent also perform better than the MSA-dependent ones. This suggests that in these cases, alignments are not very useful to score variants, possibly because of issues in these alignments. This constitutes another example of the importance of using an MSA-free but homology aware model.

---

> > ### Comment · Reviewer_biCn · 2024-11-23
> >
> > Thank you, authors, for your responses. However, I believe that both the Transformer and Transformer++ models are capable of handling such input sizes. The original Mamba paper noted: "All attention models were only tested up to a sequence length of 2^14 = 16384 due to memory limitations." So at least for first stage of 2^11 in your case, it is possible to test them. For any long-context modeling problem, I believe it is essential to benchmark against other architectures. Examples in the bioi-related field include HyenaDNA or the recently developed Evo-1.

---

> ### Author Response · Authors · 2024-11-24
>
> We thank the reviewer for their additional comments, and for suggesting a more explicit comparison to other architectures, such as transformer and Hyena-based ones. We agree that HyenaDNA and Evo-1 provide very exciting ways of addressing long context, although their objectives are quite different from those of ProtMamba (genome-level modeling versus protein family-level modeling). As the reviewer pointed out, vanilla transformers can handle sequence lengths up to 2^14 tokens in specific implementations. However, for the task of training on large protein families, which often exceed this limit, transformers still face significant computational and memory constraints.
>
> To address the reviewer’s concerns, we started training some experiments to compare ProtMamba with both a small GPT2 transformer (with Flash-Attention) and a small Hyena model (matching the size of the small Mamba model used in our ablations). The results of these experiments will be included in the final version of the paper, pending their completion.
>
> We would like to already emphasize that, whatever the results of these comparisons on limited context lengths, the limitations of transformer-based models on context lengths larger than those used in training were demonstrated in PoET, specifically in Figure 4 of [1]. We believe that their results are a strong argument to justify the use of other architectures for long context tasks.
>
> Lastly, while benchmarking against transformers for limited context lengths provides an important point of comparison, we believe that by focusing on homology-aware sequence modeling without alignment, ProtMamba addresses a pressing need for flexible, efficient models in protein science. Its utility in both predictive tasks and conditional generation highlights its versatility.
>
> [1] Truong Jr, T., & Bepler, T. (2023). Poet: A generative model of protein families as sequences-of-sequences. Advances in Neural Information Processing Systems, 36, 77379-77415. https://arxiv.org/abs/2306.06156

---

> > ### Author Response · Authors · 2024-11-29
> > **Additional comparison with Transformer and Hyena baselines.**
> >
> > Thank you again for your valuable suggestions. Below, we provide additional results about comparisons between other models (which was requested by reviewer **Ua6N** too). We will report these results in the supplementary material in later revisions.
> >
> > - **Training Setup:** We conducted a comparative evaluation of Mamba, Hyena, and GPT2 using flash attention, focusing on small-scale models, with our training framework and data. Each model was designed to have approximately 14M parameters, consisting of 8 layers with 512-dimensional representations (384 for GPT2). All models were trained with a context size of 32,000 tokens for 72 hours on single H100 GPUs. Note that we also started training a larger GPT2 model with 115M parameters. However, during the first 24 hours, the loss was decreasing much more slowly than with Mamba and Hyena, so we decided to focus on small models instead (the time necessary to converge would have been too high for the timeframe of this rebuttal).
> >
> > - **Evaluation:** We tested the perplexity of the models on a held-out validation set with two context lengths: 16,000 tokens (short context) and 128,000 tokens (long context).
> > 1. For short contexts, we report the evaluation rate (number of sequences evaluated per second) and validation perplexity (PPL) to demonstrate efficiency under standard settings.
> > 2. For long contexts, we report the evaluation rate and validation perplexity for GPT2 and Mamba. Hyena, however, could not be evaluated with context lengths beyond 32,000 tokens, as it is limited to context lengths defined during training.
> >
> > - **Comments on the comparison:** The table below shows that Hyena has a much higher perplexity than the other two architectures. As expected, GPT2 is better at modeling short contexts than Mamba but its performance decreases with longer contexts (not seen in training). Moreover, Mamba reaches a performance similar to that of GPT2 at longer contexts. We also show that GPT2 is 16- to 128-fold slower than Mamba during evaluation, which can be limiting for many applications (e.g. variant effect prediction, see the comparison with PoET in Table 1 of the paper). Furthermore, the small GPT2 model has an inference time which is still 4- to 34-fold larger than the larger ProtMamba model that we trained (107M).
> >
> > - **Future Directions with Transformer models:** The original approach (training protocol, dataset construction) presented in this paper is flexible and can be implemented with other architectures. In particular, using GPT2 seems indeed a promising direction for improving performance. However, this comes with substantially higher computational costs (in inference time in particular), particularly for longer contexts, where GPT2 is less efficient than Mamba. Another promising direction involves building on the recent emergence of hybrid models that mix state-space models and attention-based methods.
> >
> > | Model         	| Training Time | Tokens Seen | Short Context PPL (↓) | Eval Rate: Short Context (↑)  | Long Context PPL (↓) |Eval Rate: Long Context (↑) |
> > |--------------------|---------------|-------------|--------------------|--------------------------|-------------------|--------------------------|
> > | Hyena (14M)   	| 72h       	| 81.9B    	| 14.42         	| 17 $s^{-1}$    	| Fail          	| Fail                	|
> > | Mamba (14M)   	| 72h       	| 76.8B    	| 12.48         	| 33 $s^{-1}$  	| 11.89         	| 3.84 $s^{-1}$	|
> > | GPT2 (15M)    	| 72h       	| 33.3B    	| 10.68         	| 2 $s^{-1}$    	| 11.72         	| 0.03 $s^{-1}$ 	|
> > | |
> > | ProtMamba (107M)  | 400h      	| 190B     	| 9.69          	| 8.33 $s^{-1}$  	| 9.26          	| 1.04 $s^{-1}$ 	|

---

### Official Review · Reviewer_SxMF · 2024-11-02

**Soundness:** 3
**Presentation:** 3
**Contribution:** 3
**Rating:** 6
**Confidence:** 4

**Summary:**

The paper proposed a Mamba-based protein language model, ProtMamba, using concatenated sequences from protein families, with a FIM training objective. This approach allows for faster training and inference speeds. Experiments demonstrate ProtMamba’s versatility across protein fitness prediction and context-conditioned generation.

**Strengths:**

- **Novelty**: This is one of the first works to incorporate state-space models (SSMs) in protein language modeling, utilizing the Mamba architecture for efficient long-context handling.

- **Innovative Input Design**: The input consists of a concatenation of unaligned homologous sequences separated by CLS tokens, with a carefully designed masking strategy. This design effectively leverages long homology contexts, maximizing the model’s ability to capture evolutionary information. Training with a Fill-In-The-Middle (FIM) objective enables flexible application to tasks like mutational effect prediction.

- **Comprehensive Model Implementation and Training**: The authors have put substantial effort into implementing and training ProtMamba, incorporating techniques inspired by DNA modeling, such as callback mechanisms and sequence length warmup.

**Weaknesses:**

- **Performance**: ProtMamba does not show significant performance improvements over strong baselines such as ESM-2 and Tranception, which may limit its competitive edge.

- **Additional Comparisons**: Including comparisons with other baseline models, such as PoET-205M[1], SaProt[2], ProtHyena[3], or PTM-Mamba[4] would provide a fair evaluation and offer a more comprehensive view of ProtMamba’s strengths and weaknesses.

- One advantage of Mamba is its faster generation capability compared to transformer-based models. The authors could extend ProtMamba’s use cases by addressing protein sequence generative tasks, such as unconditional generation. A more detailed discussion in Section 3.4 comparing ProtMamba to other generative models would strengthen the paper. You could follow the setting and metrics in PROTEINBENCH[5] paper.


[1] Truong Jr, T., & Bepler, T. (2023). Poet: A generative model of protein families as sequences-of-sequences. Advances in Neural Information Processing Systems, 36, 77379-77415.

[2] Su, J., Han, C., Zhou, Y., Shan, J., Zhou, X., & Yuan, F. (2023). Saprot: Protein language modeling with structure-aware vocabulary. bioRxiv, 2023-10.

[3] Zhang, Y. (2024). Prothyena: A fast and efficient foundation protein language model at single amino acid resolution. bioRxiv, 2024-01.

[4] Peng, Z., Schussheim, B., & Chatterjee, P. (2024). PTM-Mamba: A PTM-aware protein language model with bidirectional gated Mamba blocks. bioRxiv.

[5] Ye, F., Zheng, Z., Xue, D., Shen, Y., Wang, L., Ma, Y., ... & Gu, Q. (2024). ProteinBench: A Holistic Evaluation of Protein Foundation Models. arXiv preprint arXiv:2409.06744.

**Questions:**

- I’m curious about the impact of the number of context sequences on ProtMamba’s performance. For example, how does performance change with 0, 5, or more sequences, and is there a threshold beyond which additional context sequences no longer contribute to model performance? In your experiments on scaling FIM perplexity with the number of context sequences, it seems perplexity stabilizes with around 30 context sequences. Could you elaborate on this?

- Inference Efficiency: Could you report on ProtMamba’s efficiency during inference? Additionally, how does ProtMamba’s performance, memory usage, and computational efficiency scale with increasing context length compared to transformer-based models, like PoET?

---

> ### Author Response · Authors · 2024-11-19
> **Response (1/1)**
>
> We thank the reviewer for their valuable feedback on our paper. We performed multiple novel ablations and comparisons that we report in the main comment of the rebuttal. We also address their comments and concerns below:
> 1. **Performance comparisons:** In Table 1, which is present both in the common answer to all reviewers and in the new version of the paper, we show that even if ProtMamba has much fewer parameters and was trained for less time, we still outperform both ESM2 and Tranception on the ProteinGym benchmark. The only models that perform better than ProtMamba are TranceptEVE (which uses an EVE model trained on the single MSAs) and PoET (which is twice larger than ProtMamba and much slower). Furthermore, all of the models we compared with are transformer-based, and therefore their inference times are 20 to 200 fold larger than those of ProtMamba. We added a column in Table 1 to compare inference times.
> 2. **Additional comparisons:** We added a comparison with PoET and SaProt in Table 1. We did not include ProtHyena and PTM-Mamba because they have a different training objective. To the best of our knowledge, they were not tested on ProteinGym because of this.
> 3. **Generative capabilities:** In the global answer to all reviewers we reported a comparison of conditional generation between ProtMamba and EvoDiff, which we believe to be the state of the art model for this task. We show that ProtMamba strongly outperforms EvoDiff. Note that we did not compare to Progen2 and DPLM because they are not designed for homolog-conditioned generation (see our main response for more details). Besides, we did not perform completely unconditional generation (without homolog information) because it is not what ProtMamba was designed for.
> 4. **Proteinbench:** We thank the reviewer for pointing out the interesting Proteinbench benchmark. We could not include it in our analysis because it was published after the ICLR deadline. Furthermore, to the best of our knowledge, the code in the ProteinBench GitHub repository is not available yet (as of 19 Nov 2024). We will be happy to test ProtMamba on Proteinbench later. Also, we benchmarked ProtMamba generative abilities against other similar models, see point 3.
> 5. **Impact of the number of context sequences on ProtMamba’s performance:** We show in Figure 2 the scaling of perplexity with the number of sequences in the context. We would like to clarify that perplexity does not stabilize after 30 sequences, but continues to decrease as a power law. To better show this, we added in the supplement the same figure in a log-log scale (the figure can be found in the pdf named: "rebuttal_attachment" in new supplementary material file).
> 6. **Efficiency and memory usage during inference:** As ProtMamba is an autoregressive recurrent model, there is no limitation in memory usage in inference.  Conversely, transformer-based language models like PoET have a limitation on context length due to memory usage.
> Regarding time complexity during inference, we show that ProtMamba has a 85-fold improvement on the time to score variants in ProteinGym with respect to PoET with no ensembling. If one uses PoET with ensembling (to reach its reported performance) then the improvement is 1300-fold. See the table below for a comparison between ProtMamba and PoET on the time taken to score all variants in ProteinGym.
> Other comparisons between Mamba-based and Transformer-based language models (like PoET) in terms of efficiency were studied in [1], which we now cite.
>
> | **Model**                 	|**Time to score variants**   |
> |-------------------------------|------------|
> | ProtMamba (single)   	|7m 	|
> | PoET (single)            	| 9h 51m 	|
> | PoET (ensemble)          	| 148h   	|
>
> [1] R. Waleffe et al. An empirical study of mamba-based language models.

---

### Official Review · Reviewer_tE4Y · 2024-11-04

**Soundness:** 3
**Presentation:** 2
**Contribution:** 3
**Rating:** 6
**Confidence:** 5

**Summary:**

The paper introduces ProtMamba, a state-space protein language model that is trained on sets of homologous sequences concatenated together. The model is trained to both generate sequences from scratch and to infill sequences using a fill-in-the-middle objective. For fitness prediction on ProteinGym, the method is shown to perform on-par with much larger models that explicitly use a multiple sequence alignment. For a narrower dataset of chorismate mutase activities, they demonstrate that they can apply prompt engineering and the FIM objective to improve fitness prediction. Finally, they perform a limited evaluation of the model’s autoregressive generation capabilities and show that the top 10% of generated sequences have some properties similar to natural proteins.

**Strengths:**

Prompting protein language models with sequences of homologous proteins (rather than an MSA) is an exciting direction for retrieval-augmented models. Given the inefficiencies of long-context transformers, using a state-space model for this objective is a natural idea to explore.

The choice to couple standard autoregressive language modeling with a fill-in-the-middle objective is an interesting one that has been relatively unexplored for protein language modeling, and the authors show its value for a fitness prediction task involving chorismate mutases.

The authors also provide a limited demonstration that prompting the model with high-activity sequences can improve its ability to perform fitness prediction.

**Weaknesses:**

Major points:
General clarity: I have a hard time following all the details of the paper because of the copious references to supplementary figures to support central claims in the main text.

Table 1: Given that the code & model parameters are publicly available, I would like to see the authors reproduce results for PoET [1], both without retrieval and with retrieval using the same prompt that is provided to ProtMamba. Given how similar the two approaches are, I cannot accept this paper without seeing this baseline.

Table 1: I would like to see how ProtMamba performs when one uses the autoregressive log likelihood of an unmasked sequence, rather than the FIM objective. Without this comparison, the value of FIM vs. autoregressive language modeling is less clear to me, since the scope of the experiment in Section 3.3 is much more limited.

Lines 208-212: More recent work [2, 3, 4] suggests that it is beneficial to mask as much as 50% of a sequence. I would like to see ablations that evaluate different masking fractions, rather than just results for the somewhat arbitrary choice of 20%.

Figure 4: There are major loss spikes and periods where the training loss actually increases. The authors should comment on the overall training stability of ProtMamba with some analysis of the gradient norms during training. This is important for a reader to decide whether they would choose to adopt Mamba over a transformer.

Minor points
Lines 89-95: Modern attention implementations like FlashAttention have linear memory complexity, though they still have quadratic time complexity [5]. The authors should update the text to reflect this fact.

Table 1: Indicating the top performers for each evaluation in bold would improve the readability of the table.

Figures 6-7: It is unclear to me what L denotes, and why the 150 < L < 250 line extends so much further than the other 2 in Figure 6.

[1] Truong Jr. & Bepler. PoET: A generative model of protein families as sequences-of-sequences. NeurIPS, 2023.
[2] Wettig et al. Should You Mask 15% in Masked Language Modeling? EACL, 2023.
[3] Tay et al. UL2: Unifying Language Learning Paradigms. arXiv, 2022.
[4] Hayes et al. Simulating 500 million years of evolution with a language model. bioRxiv, 2024.
[5] Dao. FlashAttention-2: Faster Attention with Better Parallelism and Work Partitioning. ICLR, 2023.

**Questions:**

ProteinGym’s performance metrics are computed by averaging together the Spearman correlations for all assays with the same (UniProt ID, Function) pair, computing the average-of-averages for each function, and then averaging over functions. When computing the depth-based (and other) averages, I believe the UniProt IDs are averaged first as well, though not the functions. Can the authors confirm that they use the appropriate hierarchical averages to compute results for ProtMamba in Table 1?

---

> ### Author Response · Authors · 2024-11-19
> **Response (1/2)**
>
> We thank the reviewer for their valuable feedback on our paper. We performed multiple novel ablations and comparisons that we report in the main comment of the rebuttal. We also address their comments and concerns below:
> 1. **General clarity:** Following the reviewer’s remark, we removed some non-crucial references to the supplement from the main text. We also added a new table that brings back some of the key results in supplementary in the main paper. Finally, we renumbered supplementary figures as “S1, S2…” for added clarity.
> 2. **Comparison with PoET (Table 1):** For completeness, we added the PoET official results to Table 1 which we also shared in the common response to all reviewers above and here.
> As mentioned in the main answer to all reviewers, even if PoET performs better than ProtMamba on the benchmark, an important result is that we can reach competitive performance using a fraction of the training FLOPs, a fraction of the training data and a fraction of the inference time (85 to 1300 fold reduction on the inference time on ProteinGym) as shown in the "Time" column (the time needed to evaluate all variants in ProteinGym).
> Unfortunately, it is not possible to compare the results with PoET using the exact same prompt as used in ProtMamba because of the context length limitations and inference costs of PoET.
>
> | **Model**                 	| **#Params** | **Spearman $\rho$** | **Time**   |
> |-------------------------------|-------------|------------|------------|
> | ProtMamba (w/ R)     	| 107M    	| 0.432  	| 10m    	|
> | ProtMamba (single)   	| 107M    	| 0.406  	| 7m 	|
> | PoET (single)            	| 201M    	| 0.447  	| 9h 51m 	|
> | PoET (ensemble)          	| 201M    	| 0.470  	| 148h   	|
>
> 3. **More conclusive results on protein generation:** We would like to highlight that, based on other reviewers comments, we added a comparison with other models on the conditional protein generation task (discussed in the general response to all reviewers and in section 3.4 of the updated paper) that may be of interest for the reviewer.
> 4. **Comparison between using FIM loss and autoregressive log likelihood for ProteinGym (Table 1):** We report the comparison between using the standard autoregressive log likelihood and using the FIM loss on the ProteinGym benchmark both here and in the common response to all reviewers above. We find that using FIM yields a better performance, which demonstrates its relevance for variant effect prediction. We also note that the FIM technique allows to score all the 20 possible amino-acids at each position at the same time, while the autoregressive one cannot.
>
> | **Model**                 	| **#Params** | **Spearman $\rho$** | **Time**   |
> |-------------------------------|-------------|------------|------------|
> | ProtMamba AR (single)	| 107M    	| 0.361  	| 1h 39m 	|
> | ProtMamba (single)   	| 107M    	| 0.406  	| 7m	|
>
> 5. **Choice of masking fraction:** We thank the reviewer for raising this interesting point. We chose a masking fraction of 20% in line with the majority of the literature in Masked Language Modeling that uses ~15%. Note that our model is not strictly equivalent to a model trained with MLM: as it is trained to predict residues autoregressively, both in FIM and not in FIM, it has seen a range of masking fractions. We now propose an “ablation” using 50% as a masking fraction, and obtain a minor performance improvement, as reported in the general rebuttal. We will take this into account for the new versions.

---

> > ### Author Response · Authors · 2024-11-19
> > **Response (2/2)**
> >
> > 6. **Training instabilities:** While Mamba is known to have training instabilities that cannot be fixed by clipping the gradient norm [1, 2], this problem was recently fixed in Mamba2 [3]. Consistently, our very early tests with Mamba2 do not feature such instabilities. Thus, we expect this issue to vanish in the near future. Practically, in the current version of ProtMamba, we used gradient clipping and restarted the training from a previous checkpoint to handle training instabilities (see Methods).
> > 7. **Linear memory complexity of FlashAttention:** We updated the main text according to the reviewer’s comments on FlashAttention. While the reviewer is correct that FlashAttention has a linear complexity in terms of memory, it nevertheless still has a quadratic complexity in time, which is the main bottleneck of training Transformers with respect to Mamba models. We clarified this point in the new version of the paper.
> > 8. **Table 1 formatting:** In the new version of the paper we did a major update in the format of Table 1, where we include many more methods. We used a gray background to highlight top performers, as suggested by the reviewer. We shared the new table 1 also in the common response to all reviewers.
> > 9. **Meaning of L in Figures 6-7:** L is the average length of the sequences in the family considered. We now explicitly mention this in the figures’ captions. The 150 < L < 250 line extends more than others because the distribution of cluster sizes is not uniform. Those clusters with L<150 tend to be more shallow (fewer homologs) than those with 150<L<250, leading to a concatenated length smaller than the maximum context length (131k tokens); instead, for L>250 the limit on the context length (131k tokens) was reached when concatenating sequences. This is the reason why the curves with intermediate lengths (150<L<250) are longer than the other cases.
> > 10. **Method used to compute the scores on ProteinGym:** We confirm that we used the same hierarchical method of computing the averages of the Spearman correlation scores as detailed in the ProteinGym paper. We added a section in the Supplement where we describe the methodology to score the variants. We checked that for known models, we find the same Spearman correlation as the one reported in the ProteinGym benchmark.
> >
> > [1] R. Waleffe et al. An empirical study of mamba-based language models.
> >
> > [2] E. Nguyen et al. Sequence modeling and design from molecular to genome scale with Evo.
> >
> > [3] T. Dao et al. Transformers are SSMs: Generalized Models and Efficient Algorithms Through Structured State Space Duality

---

> ### Author Response · Authors · 2024-11-25
> **Additional comparison with PoET on shorter prompts**
>
> We would also like to add a comparison between ProtMamba and PoET where we use exactly the same prompt, i.e. using a short context of 12k amino acid tokens. We obtain:
>
> | **Model**                 	| **#params** | **Spearman ρ**    	| **Time**    	|
> |-------------------------------|-------------|--------------|-----------------|
> | ProtMamba (12k)   	| 107M    	| 0.381   	| 7m        	|
> | ProtMamba (single)   	| 107M    	| 0.406    	| 7m        	|
> | PoET (single)           	| 201M    	| 0.447    	| 9h 51m      	|
>
> This result shows that ProtMamba is not as good as a larger transformer model when using a short context length. It also shows that leveraging more homologs is very beneficial for the model’s performance on this task. More detailed information  on how the context length impacts the performance of ProtMamba is included in figures S6, S7 and S8 in the updated paper.

---

> > ### Comment · Reviewer_tE4Y · 2024-11-26
> >
> > I have read the authors' response and updated manuscript, and am happy to increase my score in light of the additional analyses performed by the authors.

---

### Author Response · Authors · 2024-11-19
**Response to all reviewers (1/2)**

We thank the reviewers for their valuable feedback on our paper. We addressed all their questions with official comments on the individual reviews. We report here a brief summary of the main analyses and ablation that we performed in response and that we added to the new version of the paper. We provided a revised version of the manuscript with changes highlighted in blue.

1. **Ablations on the training regime:** We trained different models (14M parameters d_model=512, n_layers=8 and 107M parameters like the original ProtMamba) for N=50k steps, reaching a total number of training tokens of T=10B. We report the perplexity computed on the validation set in the following table that we added to the new version of the paper. We signal with "Fail" the ablations where the model fails (e.g. a model trained with just the FIM loss has a very high AR loss on the full sequence).

| **Perplexity**          | **14M Parameters**    	|            	| | | **107M Parameters**    	|            	|
|---|---|---|-|-|---|---|
|                          	| **Autoregressive**   	| **FIM**    	| | | **Autoregressive**     	| **FIM**    	|
||
| Only FIM from scratch    	| Fail                 	| 13.90 ± 0.34   | | | Fail                   	| 15.59 ± 0.27   |
| AR only                  	| **12.58** ± 0.31     	| 18.03 ± 0.25   | | | 11.05 ± 0.36           	| Fail       	|
| No positional encoding    	| 13.01 ± 0.30        	| 16.71 ± 0.47   | | | 12.31 ± 0.37           	| 17.20 ± 0.58   |
| Additive positional encoding  | 12.72 ± 0.31        	| 13.60 ± 0.33   | | | 12.58 ± 0.38           	| 13.81 ± 0.31   |
| One mask, one token       	| 12.76 ± 0.31        	| 15.54 ± 0.29   | | | 11.04 ± 0.33           	| 16.60 ± 0.36   |
| Masking fraction 50%      	| 13.02 ± 0.31        	| **13.44** ± 0.33 | | | **10.94** ± 0.36       	| **11.59** ± 0.35 |
||
| ProtMamba                	| 13.00 ± 0.30         	| 13.89 ± 0.32   | | | 11.35 ± 0.33           	| 12.62 ± 0.30   |

Main observations from the ablations:
- Raising the masking fraction to 50% has a minor positive effect on the model, we will take this into account in future versions.
- The use of positional embeddings does not deteriorate performance on the AR loss, it actually improves it on the FIM loss allowing us to perform inpainting with length control during inference.
- There is no substantial difference between sum and concatenation of the positional embeddings in small models. In large models, concatenation is slightly better.
- Purely autoregressive training hinders the FIM capabilities of the model.
- Using one mask per token degrades the FIM capabilities of the model (which are needed when one wants to inpaint a sequence).
- We are still performing ablations on a vanilla GPT2 Transformer.

2. **Generative abilities and comparison with other state of the art models:** We compare the generative capabilities of ProtMamba with other state of the art models for conditioned generation, namely EvoDiff-MSA, MSA-Transformer and Potts models. Specifically, we now report a comparison of the pLDDT (using ESMFold) and scPerplexity (using ProteinMPNN) of 250 novel protein sequences generated using ProtMamba, each from a different cluster in our test set. We compare these values to the same scores measured on 250 novel protein sequences generated by EvoDiff-msa, MSA-Transformer and Potts models, retrieved from the Zenodo archive associated to the EvoDiff paper [1], and which were generated each from a different cluster of the EvoDiff validation set. We find that ProtMamba strongly outperforms all these models on conditioned generation, and obtains scores comparable to those of natural sequences, see table below.

| Model           | ProtMamba   |   	| EvoDiff   |   	| MSA Transformer	|   	| Potts    	|   	| Natural   |
|---|---|---|---|---|---|---|---|---|---|
| pLDDT (↑) | **0.75 ± 0.13** |   	| 0.60 ± 0.16   |   	| 0.54 ± 0.18   |   	| 0.56 ± 0.14   |   	| 0.77 ± 0.13   |
| scPerplexity (↓) | **2.63 ± 0.45** |   	| 3.17 ± 0.58  |   	| 3.37 ± 0.64  |   	| 3.17 ± 0.51  |   	| 2.66 ± 0.49   |

Some observations on other known models that we did not include in this comparison:
- We did not perform this analysis using PoET because the authors did not release the code to sample from their model (they did share a script to score the variants).
- We did not perform this analysis using other state of the art de novo generative models like ProGen2 or DPLM because it is not possible to generate novel sequences conditioned on homologs using them, contrary to ProtMamba and EvoDiff. For this reason, we considered EvoDiff as the state of the art model for conditional generation. Conceptually, it is tempting to compare ProGen2’s control tag-conditioned generation or DPLM2’s structure-conditioned generation to ProtMamba’s homolog-conditioned generation. However, while these approaches are related, quantitative comparisons would be challenging because the exact conditioning would differ.

---

> ### Author Response · Authors · 2024-11-19
> **Response to all reviewers (2/2)**
>
> 3. **Comparison with other models on the ProteinGym benchmark:** We report a more detailed comparison with PoET and other models on the ProteinGym benchmark, together with a comparison of inference time and number of parameters between PoET and ProtMamba. Unfortunately, this comparison is partially incomplete as the training time and FLOPs for PoET are not available.
> We furthermore add a comparison on the inference time of PoET and ProtMamba, showing that ProtMamba has a 85-fold improvement with respect to PoET (single) and a 1300-fold improvement with respect to PoET (ensemble) on the complete ProteinGym benchmark.
>
> | **Model type**       | | **Model**                     | | **#params** | | **Spearman $\rho$** | | **Time**        |
> |-----------------------|-|-------------------------------|-|-------------|-|-----------------|-|-----------------|
> | **Alignment-based**   | | Site-Independent             | | -           | | 0.359          | | -               |
> |                       | | GEMME                        | | -           | | 0.455          | | -               |
> |                       |
> | **PLM**              | | Tranception L (w/o R)        | | 700M        | | 0.374          | | -               |
> |                       | | ESM-2                        | | 150M        | | 0.387          | | -               |
> |                       | | ESM-2                        | | 650M        | | 0.414          | | -               |
> |                       |
> | **Homology-aware**    | | ProtMamba (single)           | | 107M        | | 0.406          | | **7m**          |
> | **PLM**              | | ProtMamba AR (single)        | | 107M        | | 0.367          | | 1h 39m          |
> |                       | | PoET (single)               | | 201M        | | 0.447          | | 9h 51m          |
> |                       | | PoET (ensemble)             | | 201M        | | 0.470          | | 148h           |
> |                       |
> | **Alignment + PLM**   | | ProtMamba (w/ R)             | | 107M        | | 0.432          | | 10m             |
> |                       | | MSA-Transformer             | | 100M        | | 0.421          | | -               |
> |                       | | Tranception L (w/ R)        | | 700M        | | 0.434          | | -               |
> |                       | | VespaG                      | | 3B          | | 0.458          | | -               |
> |                       |
> | **Structure-aware**   | | ESM-IF1                     | | 142M        | | 0.422          | | -               |
> |                       | | SaProt                      | | 650M        | | 0.457          | | -               |
> |                       | | ProSST                      | | 110M        | | **0.507**      | | -               |
>
> [1] Sarah Alamdari et al. Protein generation with evolutionary diffusion: sequence is all you need.

---

### Meta-Review · Area_Chair_8Dox · 2024-12-21

**Metareview:**

This paper introduces a "homology-aware" but "alignment-free" model called ProtMamba that combines protein language model and the Mamba architecture.
Instead of directly utilizing multiple sequence alignment (MSA), ProtMamba leverages protein language model to leverage individual homologous protein sequences without the need for their alignment.
The reviewers commend this is an interesting research direction with some novelty.
However, ProtMamba doesn't always lead to significant performance improvement and does not convincingly demonstrate the advantage of the proposed scheme over other existing approaches, some of them with smaller size and utilize traditional attention-based approach.
Overall, while the work is promising, additional benchmarks against other existing schemes and providing further rationale and justification regarding the advantages of the proposed model architecture over other existing alternatives (esp., smaller and simpler models)  would be needed to further strengthen the current work.

**Additional Comments On Reviewer Discussion:**

The authors have actively engaged with the reviewers during the discussion period.
The authors have provided additional experimental results and explanations that have addressed the initial concerns of the reviewers to some extent.
Most reviewers have responded to the authors and engaged in the discussion.
The AC finds that the additional evidence provided by the authors is useful, but generally agrees with the reviewers regarding the need for additional experiments, comparison, and discussions/analysis to strengthen the manuscript.

---

### Decision · Program_Chairs · 2025-01-22

Reject